

# An overview of cloud-radiation denial experiments for the Energy Exascale Earth System Model version 1

Bryce E. Harrop[1], Jian Lu[1], L. Ruby Leung[1], William K. M. Lau[2], Kyu-Myong Kim[3], Brian Medeiros[4], Brian J. Soden[5], Gabriel A. Vecchi[6,7], Bosong Zhang[8,9], and Balwinder Singh[1]

[1]Pacific Northwest National Laboratory, Richland, WA, USA
[2]Earth System Science Interdisciplinary Center, University of Maryland, College Park, MD, USA
[3]Climate and Radiation Laboratory, NASA/Goddard Space Flight Center, Greenbelt, MD, USA
[4]National Center for Atmospheric Research, Boulder, CO, USA
[5]Rosenstiel School of Marine and Atmospheric Science, University of Miami, Miami, FL, USA
[6]Department of Geosciences, Princeton University, Princeton, NJ, USA
[7]5High Meadows Environmental Institute, Princeton University, Princeton, NJ, USA
[8]Program in Atmospheric and Oceanic Sciences, Princeton University, Princeton, NJ, USA
[9]NOAA/Geophysical Fluid Dynamics Laboratory, Princeton, NJ, USA

**Correspondence:** Bryce E. Harrop (bryce.harrop@pnnl.gov)

**Abstract.** The interaction of clouds and radiation is a key process within the climate system, and assessing the impacts of that interaction provides valuable insights into both the present day climate and future projections. Many modeling experiments have been designed over the years to probe the impact of the cloud radiative effect (CRE) on the climate, including those that seek to disrupt the mean CRE effect and those that only disrupt the covariance of the CRE with the circulation. Seven

such experimental designs have been added into the U.S. DOE's Energy Exascale Earth System Model version 1 (E3SMv1). These experiments include both the first and second iterations of the Clouds On-Off Klimate Intercomparison Experiment (COOKIE) experimental design, as well as the cloud-locking method. This manuscript documents the code changes necessary for implement such experiments and also provides detailed instructions for how to run them. Analyses across experiment types provide valuable insights and confirm the findings of prior studies, including the role of cloud-radiative heating toward

intensifying the monsoon, intensifying rain rates, and poleward expansion of the general circulation owing to cloud feedbacks.

## 1 Introduction

The interaction of radiation with clouds, termed the Cloud Radiative Effect (CRE), is a strong modulator of the global energy budget. The different ways clouds interact with shortwave (SW) and longwave (LW) radiation create two very different impacts

for the climate. The SWCRE cools the Earth by reflecting sunlight back to space, while the LWCRE heats the Earth by absorbing LW. Globally, SWCRE exceeds LWCRE (Loeb et al., 2018), but over the deep convective regions of the Tropics, the SWCRE and LWCRE are much closer to each other in magnitude owing to offsetting SW and LW CRE from tropical upper-



level clouds (Ramanathan et al., 1989; Kiehl, 1994; Tian and Ramanathan, 2002). This offset does not, however, imply that clouds have no impact because the SWCRE primarily cools the surface while the LWCRE primarily warms the atmosphere.

The surface cooling and atmospheric warming act as an indirect energy transfer from the surface to the atmosphere (Tian et al., 2001), and this energetic pathway amplifies the energy export of the Tropics (Tian et al., 2001; Harrop and Hartmann, 2015). CREs have also been shown to modify tropical circulation patterns (Harrop and Hartmann, 2016; Popp and Silvers, 2017; Albern et al., 2018; Fläschner et al., 2018; Benedict et al., 2020) and extratropical circulation patterns (Ceppi et al., 2012; Li et al., 2015; Watt-Meyer and Frierson, 2017; Chen et al., 2021).

Voigt et al. (2021) note that while much research has been done concerning the role of the atmospheric CRE (ACRE) and its impact on circulations, many of the studies testing the impact of ACRE have relied on zonally symmetric aqua planet simulations with prescribed sea surface temperatures. Voigt et al. (2021) advocate for additional research efforts to better understand the impact of ACRE on circulations with realistic boundary conditions including realistic land configurations and a careful examination of regional impacts.

Numerous methods for examining the role of ACRE on circulations have been proposed and implemented into models. For example, the Clouds On/Off Klimate Intercomparison Experiment (COOKIE; Stevens et al., 2012) removed clouds from the radiative transfer process in climate model simulations to study their impacts. When land was included, however, dramatic changes in the land surface fluxes occurred owing to the reduced cloud shading (increased surface SW fluxes) at the land surface (Webb et al., 2017). As a result, for a follow up set of experiments, COOKIE2, part of the Cloud Feedback Model

Intercomparison Project (CFMIP) contribution to Coupled Model Intercomparison Project Phase 6 (CMIP6; Eyring et al., 2016), it was suggested that only the LW portion of CRE be turned off in COOKIE-style experiments to allow realistic boundary conditions to be used without the severe land-sea temperature contrast changes (Webb et al., 2017). An alternative method was proposed by Aiko Voigt where clear-sky heating rates are applied in the atmosphere, but the surface still "sees" the all-sky fluxes (Webb et al., 2017). At the time, this method was not selected as no pilot study had yet been accomplished showing

its efficacy. A subsequent study by Dixit et al. (2018) demonstrated the method for the aquaplanet configuration, but results have not been shown for realistic geography to test whether the all-sky versus clear-sky surface flux distinction is effective. All three of these methods are designed to remove the entirety of the ACRE signal, but there exists an alternative framework that is designed to disrupt the interactions of clouds with the climate, while holding the mean CRE fixed. One such way of accomplishing this is through cloud-locking — where the model cloud optical properties are replaced by values taken from a

control simulation. Voigt and Albern (2019) note that COOKIE-style experiments (those experiments that remove the mean CRE) are valuable for understanding the present-day climate and its response to ACRE, but for climate change, cloud-locking experiments are needed (Voigt and Albern, 2019). The cloud-locking method has also been shown to be useful for quantifying the cloud radiative component of changes in SSTs under forcing (Trossman et al., 2016; Middlemas et al., 2019, 2020; Chalmers et al., 2022; Hsiao et al., 2022; Boehm and Thompson, 2023).

The goals of this manuscript are: (i) to document the modifications made to E3SMv1 to run a variety of CRE-denial experiments (including multiple experiments removing the mean ACRE as well as multiple experiments removing only the covariance between ACRE and circulation); (ii) provide detailed descriptions of how to setup and run these types of experiments on HPC



systems; and (iii) to provide a few examples of quantitative assessments of the impact of ACRE on the water cycle and cir-
culation. By documenting each of these experiments, we hope to shed light on the value and limitations of each for better
understanding the role of cloud radiative interactions on climate processes.

## 2   Model description

We make use of the Energy Exascale Earth System Model v1 (E3SMv1; Golaz et al., 2019). The cloud radiation modifications
are made to the E3SM Atmosphere Model (EAM; Rasch et al., 2019). EAM uses a spectral element dynamical core (Dennis
et al., 2012) for solving the primitive equations, the Cloud Layer Unified by Binormals parameterization (CLUBB; Larson et al.,
2002; Golaz et al., 2002a, b) for modeling shallow convection, macrophysics, and turbulence, the Morrison-Gettelman version
2 microphysics parameterization (Gettelman and Morrison, 2015; Morrison and Gettelman, 2008) for cloud microphysics, the
Zhang-McFarlane deep convective parameterization (Zhang and McFarlane, 1995) with modifications by Neale et al. (2008)
for modeling deep convection, the four-mode version of the Modal Aerosol Module parameterization (MAM4; Liu et al., 2016)
for modeling aerosols, and the Rapid Radiative Transfer Model for general circulation models (RRTMG; Iacono et al., 2008;
Mlawer et al., 1997) for the radiative transfer. We use a set of tunings described by Ma et al. (2022) that improve upon the
simulated climate of E3SMv1 (see Ma et al., 2022, for more details). The exact values of each tuning change from the default
E3SMv1 can be found in the sample run script contained within the supplementary materials.

All experiments are run as "AMIP" style experiments with realistic land-sea geography, an active land model, and prescribed
sea surface temperatures (SSTs) and sea-ice concentrations. The control configuration uses repeating SSTs and sea-ice concen-
trations based on a 20-year monthly climatology centered on year 2000 of the real world. The annually repeating SST pattern
means there is no interannual variability, including ENSO, in any of the results presented in this manuscript. The model is
spun-up for 30 years to ensure the global soil moisture is in equilibrium prior to running the control simulation. We then run
the control and all experiments based on those initial conditions taken from the end of that spin-up experiment.

In addition to the control, an experiment was conducted in which a uniform warming of 4 K added to the SSTs at all points
in space and time. The spin-up procedure described above is repeated for the +4K experiment to ensure soil moisture has a
chance to come into a new equilibrium with the change in SSTs.

## 3   Experiment descriptions

We have performed a variety of experiments to disable some or all impacts of CRE on the climate. The experiments can broadly
be grouped into two separate categories. The first is a complete denial of the cloud radiative heating (the "COOKIE-style" type
of experiment). We use the term "complete" here to refer to the removal of both the mean cloud radiative heating and its
covariance with the atmospheric circulation. These experiments include the original COOKIE experiment and its successor
described above, as well as two additional variants which we describe in detail in the following subsections. The second group
denies the covariance of CRE and circulations while maintaining the mean CRE. By maintaining the mean cloud radiative




heating, this second category seeks to preserve the general circulation pattern of the control simulation. Preserving the general
circulation relies on the assumption that the covariance term has a much smaller impact than the mean, which has been shown
to be the case for several models and metrics (Voigt and Albern, 2019). We perform multiple simulations in each of these
two categories. The variations in each experiment allow us to make more nuanced evaluations of aspects of the impact cloud
radiation interactions have on the climate state.

## 3.1 Complete cloud radiative effect denial

The original COOKIE experiment described by Stevens et al. (2012) is designed to completely remove all impacts of cloud-
radiation interactions from the simulation. While the COOKIE experiment design can be useful for assessing the total impact
of CREs, with some modifications it has also been shown to be valuable for testing the impact of different cloud types (e.g.
Fermepin and Bony, 2014; Dixit et al., 2018) or separating the impacts from LW and SW CREs (e.g. Popp and Silvers, 2017).
In this spirit, we have designed four experiments which we describe in detail in the following subsections.

Before describing the individual experiments, however, it is informative to describe the modifications made to the E3SMv1
source code to enable the removal of cloud-radiation interactions. As noted by Stevens et al. (2012), there are two ways to
accomplish this task. The first method is to make clouds transparent to every radiative transfer call done within the model (e.g.
setting cloud optical depth to zero everywhere). The other, more complicated, method is to replace the all-sky fluxes with their
clear-sky values. The latter approach is used for E3SMv1. While more challenging, it offers a pair of benefits. First, as noted
by Stevens et al. (2012), it allows for the normal model outputs to be used to assess how the clouds respond to decoupling from
their radiative heating. Second, the impact of clouds can be turned off individually for the surface and atmosphere.

To accomplish the latter benefit of separating the surface and atmospheric cloud radiative effects, we implement four flags
to the source code:

```
(i)    no_cloud_lw_radheat_atm
(ii)   no_cloud_sw_radheat_atm
(iii)  no_cloud_lw_radheat_sfc
(iv)   no_cloud_sw_radheat_sfc
```

The first two of these input flags control whether the all-sky (including clouds) or clear-sky (cloud-free) fluxes are used to
compute the radiative heating within the atmosphere (for LW and SW, respectively). The third and fourth of these input flags
control whether the all-sky or clear-sky fluxes are used at the surface (again, for LW and SW, respectively). By using four
flags to control the model behavior, the user has flexibility to control LW and SW CREs independently (which is valuable for
the experiment described in section 3.1.2) as well as the flexibility to treat the atmosphere and surface independently (which
is valuable for the experiment described in section 3.1.3. Each flag is set to false by default, which results in normal model
behavior, with the cloud radiative heating included in the atmospheric and surface temperature tendencies. More precisely,
when running with the default flag settings, the model is bit-for-bit (BFB) identical to simulations run with the same code base
prior to these flags, and the code they control, being implemented. For simplicity, we order the flags (i-iv) above and use 'T' or



**Table 1.** A list of the experiments used in this study, what type they belong to, and the prescribed SSTs used. A prescribed SST type of "mix-and-match" refers to the T0C1 and T1C0 scenarios used for the cloud-locking, prescribed-RadHt, and prescribed-CRE experiments.

| Experiment name | Experiment type | SSTs prescribed | Flag settings (i, ii, iii, iv) |
|---|---|---|---|
| Control | control | present-day, +4K | FFFF |
| Clouds-off | complete radiation denial | present-day, +4K | TTTT |
| Clouds-off LW | complete radiation denial | present-day, +4K | TFTF |
| Clouds-off ATM | complete radiation denial | present-day, +4K | TTFF |
| Surface-locking | complete radiation denial | present-day, +4K | TTFF |
| Cloud-locking | CRE-circulation decorrelation | present-day, +4K, mix-and-match | FFFF |
| Prescribed-RadHt | CRE-circulation decorrelation | present-day, +4K, mix-and-match | FFFF |
| Prescribed-CRE | CRE-circulation decorrelation | present-day, +4K, mix-and-match | FFFF |

'F' to denote whether a flag is set to true or false. For the control experiment, the flag settings are abbreviated as FFFF. If, for example, only the atmospheric heating flags were set to true, such a configuration would be abbreviated TTFF.

Figure 1 details how the "`no_cloud`" flags modify the EAMv1 source code. All of the changes are contained within the `radiation_tend` subroutine of the EAMv1 `radiation.F90` module (part of RRTMG). The call to compute SW fluxes is impacted by whether `no_cloud_sw_radheat_sfc` is set to true or false. The land model requires SW surface fluxes separated into four components: direct visible ($< 700$ nm), diffuse visible, direct near-infrared ($\geq 700$ nm), and diffuse near-infrared. The SW flux calculation provides these components, but only for their all-sky values. Not having the clear-sky values for these four parts of the surface SW flux means we cannot simply swap out the all-sky for the clear-sky values as desired. Note that for the net SW flux computed by RRTMG, both all-sky and clear-sky fluxes are returned and simply interchanging which value is provided to the coupler (the ocean model requests the net SW flux) is easy to do and is demonstrated in Figure 1. Thus, the only way to get the clear-sky values of the surface SW flux components needed for the land model — without more intrusive changes to the underlying RRTMG source code and its interface with EAMv1 — is to call the SW flux calculation twice. While we could still pass zero cloud optical depth to the radiation call, we want to keep the model outputs consistent across all experiments (where the all-sky flux outputs are the values of what they would be with the model produced clouds). The first SW flux calculation removes clouds (by specifying the cloud optical depth to be identically zero in all grid cells) and sets the separated surface SW flux components to be passed to the land model. The second call specifies the clouds as normal, but the output variables for the separated surface SW flux components are written to dummy variables that are discarded. The net effect of these two calls to the SW flux calculation provides the separated surface SW flux components for clear-sky conditions, while all remaining fluxes are otherwise the same. These remaining fluxes include the diagnostic values written to the atmospheric history files (e.g. net surface SW, total downwelling SW irradiance). The logic for the SW flux calculation is





depicted in the first blue box in Figure 1. For the LW flux calculations, no changes to how the LW flux calculation is called are needed to accommodate the flags (first orange box in Figure 1).

Unlike the land model, the ocean component model receives the total (sum of all four components) of SW fluxes. These net
SW flux values provided to the ocean are overwritten with their clear-sky fluxes after the diagnostic values have been written out to the history files, like what is done for the surface LW, atmospheric SW, and atmospheric LW fluxes (described below).

The history outputs are stored immediately after the fluxes are computed such that the variables include information about the clouds. For example, the CREs written to file (SWCRE is called `SWCF` and LWCRE is called `LWCF` in EAM history files) will be non-zero even when the model temperature tendency is updated using the clear-sky heating rates. One important impact
of choosing to output the computed all-sky values is constructing an energy budget for the atmosphere requires the user to select the appropriate boundary fluxes (all-sky or clear-sky) depending on how the flags have been set. In EAM, this looks like the following

```
NET_RAD = FSNT  − FLNT  − FSNS  + FLNS  ! when flags are all false
NET_RAD = FSNTC − FLNTC − FSNSC + FLNSC ! when flags are all true
```

where `FSNT` is the net top-of-model SW flux, `FLNT` is the net top-of-model LW flux, `FSNS` is the net surface SW flux, `FLNS` is the net surface LW flux, and a 'C' at the end denotes the clear-sky value. For other combinations (e.g., only `no_cloud_lw_radheat_atm` and `no_cloud_lw_radheat_sfc` are set true) a mix of all-sky and clear-sky fluxes are needed to compute `NET_RAD`.

After the history outputs have been saved (second blue and orange boxes in Figure 1), the CRE flags introduced above are
then used to control whether the clear-sky values will overwrite the all-sky values or not (third and fourth blue and orange boxes in Figure 1). After all four flags are evaluated and the surface fluxes and layer heating rates potentially overwritten, then the temperature tendency will be updated. The radiative heating is applied as a temperature tendency multiplied by the time step length, and the surface LW and SW fluxes are passed to the coupler for the surface components' boundary conditions during the next model time step. Again, Figure 1 summarizes these code modifications in a flow chart. The following sections
describe how these flags have been toggled to produce several variations of COOKIE-style experiments.

### 3.1.1 Clouds-off

The "clouds-off" experiment repeats the original COOKIE method of turning off clouds to radiative transfer for both the atmosphere and surface (Stevens et al., 2012). To run the E3SMv1 clouds-off experiment the following flag configuration is set: TTTT.
If the model is run as an aquaplanet with prescribed SSTs, then the `no_cloud_lw_radheat_sfc` and `no_cloud_sw_radheat_sfc` flags become irrelevant. For model configurations that include active land, ocean, or sea-ice components, then these "`_sfc`" flags are necessary. The clouds-off experiment is run with an active land component and prescribed SSTs and sea-ice concentrations (as are all of the simulations documented in this manuscript). Thus, while the SSTs remain the same across experiments, the land surface temperatures are allowed to respond to changes in cloud radiative fluxes.



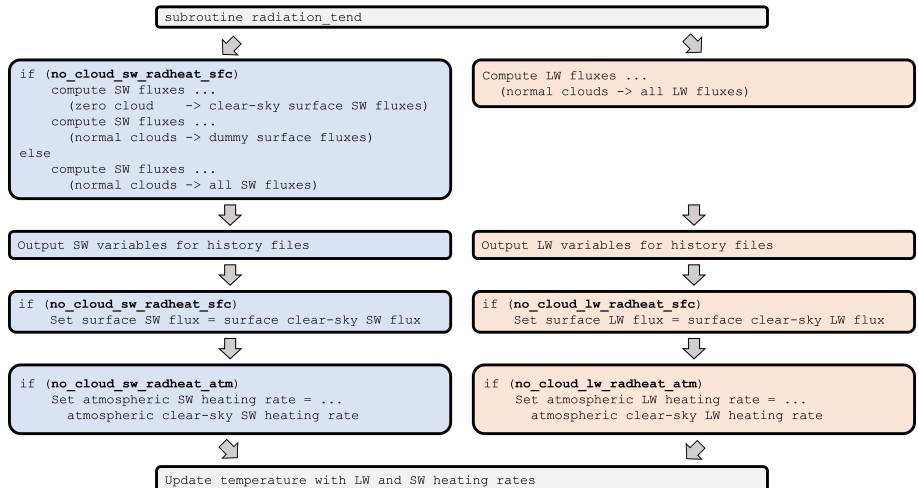

**Figure 1.** Diagram denoting the code changes controlled by the `no_cloud_lw_radheat_atm`, `no_cloud_sw_radheat_atm`, `no_cloud_lw_radheat_sfc`, and `no_cloud_sw_radheat_sfc` flags. The flags have been given in bold-face font to make their occurrence easier to identify. The blue highlights denote modifications impacting SW fluxes and the orange highlights denote modifications impacting LW fluxes. The gray arrows denote the order in which these calls are made within EAM's `radiation_tend` subroutine. The SW and LW calculations do not happen in parallel, but they are independent of one another, so they are shown side-by-side in this schematic for illustrative convenience.

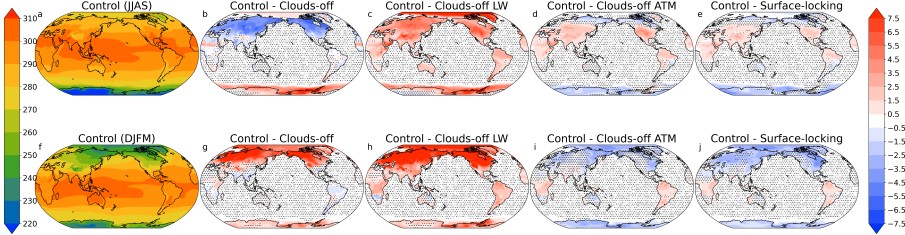

**Figure 2.** Surface temperature for Boreal summer (JJAS; top row) and winter (DJFM; bottom row) for the control experiment (a and f). Differences between the control and the various complete cloud radiative effect denial experiments (control minus experiment) are provided in the other panels. Stippling denotes locations where differences are not statistically significant at the 95th percentile. All values are shown with units of Kelvin.

Like in the original COOKIE simulations, the lack of cloud shading leads to changes in the land surface temperature (Webb et al., 2017). The surface temperature differences are largest during summer and winter with a large amount of cancellation for the annual average (see Figure 2b and g). In JJAS, SW fluxes dominate the surface energy flux over Northern Hemisphere land, allowing the surface SWCRE to cause the significant cooling seen in Figure 2b (see also Table 2). In DJFM, surface LWCRE warms the surface, giving rise to the large warming seen in Figure 2g (see also Table 2). Table 2 shows that clouds cause a



**Table 2.** Surface fluxes for the control and all experiments. The fluxes are net shortwave, net longwave, and total (sensible plus latent) heat. All flux values are positive downwards. The radiative fluxes are either the all-sky or clear-sky depending on whether the surface "sees" the clouds or not. For the experiments, the fluxes are listed as differences (control minus experiment) to highlight the impact of clouds on that experiment. For experiments like Clouds-off ATM and Cloud-locking these values are small, as expected, because the surface "sees" the clouds in these experiments. Northern Hemisphere land refers to model grid cells where the land fraction exceed 50% and are northward of 30°N.

| N. Hem. Land | JJAS | | | DJFM | | |
|---|---|---|---|---|---|---|
| | SW flux | LW flux | TH flux | SW flux | LW flux | TH flux |
| Control | 146.1 | -63.2 | -74.4 | 66.3 | -48.7 | -23.2 |
| Clouds-off | -42.5 | 20.2 | 19.5 | -16.9 | 21.4 | -3.7 |
| Clouds-off LW | -1.7 | 17.9 | -14.8 | -0.9 | 16.9 | -13.9 |
| Clouds-off ATM | -0.7 | 1.8 | -0.8 | -3.0 | 3.8 | -1.4 |
| Surface-locking | -1.3 | 2.2 | -0.3 | -1.5 | -1.2 | -0.5 |
| Cloud-locking | 6.0 | -3.2 | -2.7 | 3.5 | -3.5 | -0.0 |
| Prescribed-RadHt | 2.7 | 0.0 | -2.5 | -0.7 | 2.5 | -1.2 |
| Prescribed-CRE | 0.8 | 0.9 | -1.4 | -1.4 | 2.6 | -0.8 |

significant reduction in surface total heat flux relative to the control (primarily a reduction in evapotranspiration, not shown), in response to the large reduction in downwelling surface shortwave fluxes.

### 3.1.2 Clouds-off LW

The cloud radiation denial experiment for the CFMIP contribution to CMIP6 (the next generation of COOKIE experiments Webb et al., 2017) removes only the cloud-LW interactions. To run such an experiment with E3SMv1, the flag setting is TFTF.

We refer to this experiment as "clouds-off LW." The clouds-off LW experiment still allows SW all-sky fluxes to update the model tendency terms. While ACRE has only a small contribution from SW (see Figure 3c), SW dominates the surface CRE across much of the globe, and so it was expected that the differences in surface temperature between the clouds-off LW and control experiments would be less than those between the clouds-off and control experiments. For high-latitudes, however, the surface LWCRE is important, with the net CRE becoming positive poleward of 60° (Figure 3b). Like in the clouds-off

experiment, the surface LWCRE is removed for the clouds-off LW experiment. As a result, there are still significant surface temperature differences between the control and clouds-off LW experiment (see Figure 2c and h). There is little seasonal difference in the temperature response in the clouds-off LW experiment, because the surface LWCRE is similar throughout the year (not shown). The surface LWCRE is always positive because the emissivity of clouds is larger than that of the clear-sky, so clouds emit more downwelling LW than surrounding clear-sky regions and heat the surface (Slingo and Slingo, 1988).




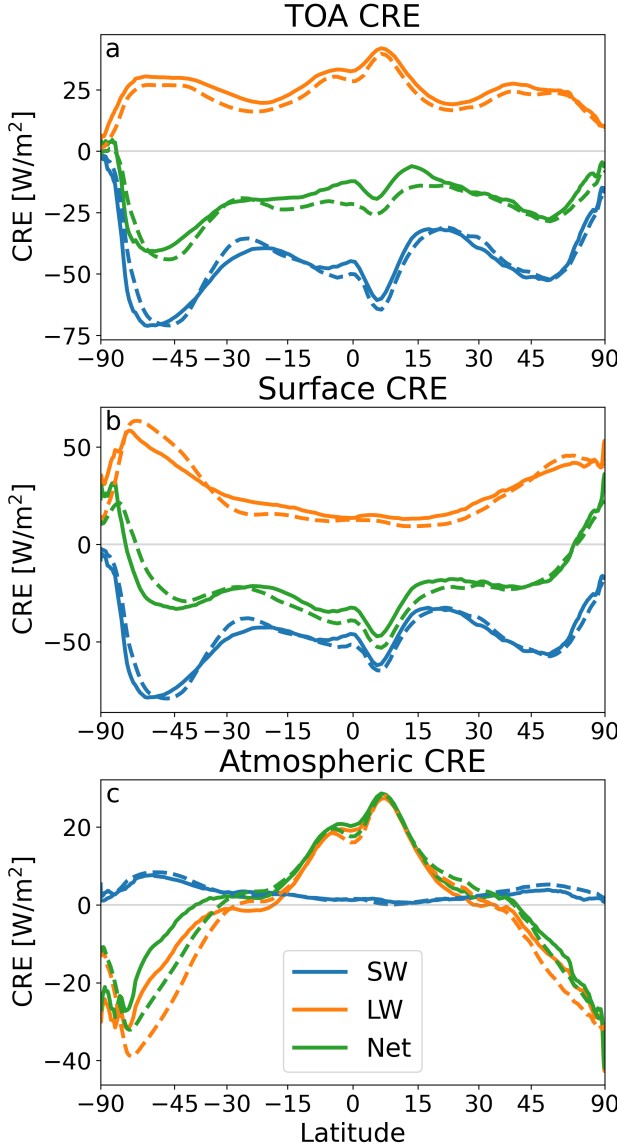

**Figure 3.** Annual mean, zonal mean Cloud Radiative Effect (CRE) at the (a) top-of-atmosphere, (b) surface, and (c) in the atmosphere. The solid lines use CERES-EBAFv4.1 data (NASA/LARC/SD/ASDC, 2019) and the dashed lines are from the control simulation. The blue line denotes SW, the orange line denotes LW, and the green line denotes the Net flux. All values are given in $\mathrm{Wm^{-2}}$.

The temperature differences relative to the control for the clouds-off LW experiment are of similar magnitude to those in the clouds-off, meaning some of the surface temperature biases this experiment is designed to alleviate are still present.

     The next generation of COOKIE experiments run for the CFMIP contribution to CMIP6 (COOKIE2, for short) show similar warming patterns (Figure 4). These COOKIE2 experiments rely on the "amip" and "amip-lwoff" experiments, meaning that





they also use prescribed SSTs, but the SSTs are transient and follow the observed evolution instead of repeating the same

pattern each year. Figure 4 shows this difference in SST prescription has no qualitative effect on the warming pattern resulting from the surface LWCRE. This robust land warming from surface LWCREs suggests caution should be exercised when using the clouds-off LW or the COOKIE2 experiments for extratropical analyses that are sensitive to land temperature.

### 3.1.3 Clouds-off ATM

An alternative to removing only the cloud-LW interactions was proposed by Aiko Voigt and considered for the CFMIP contri-

bution to CMIP6 (refer to discussion in Webb et al., 2017). This alternative experiment removes only the atmospheric heating from cloud-radiation interactions, while maintaining the all-sky surface fluxes. To run this "clouds-off ATM" experiment, the flag setting is TTFF. The clouds-off ATM configuration only removes the ACRE while still allowing surface CREs to contribute to the surface tendencies. It is important to note that the cloud shading of surface SW or increased downwelling LW are generated by the simulated clouds within the clouds-off ATM. In other words, if the cloud fields shift in location, the surface

CREs shift too. By allowing the surface to "see" the clouds, the temperature drift is significantly reduced in the control vs clouds-off ATM relative to the differences produced by the clouds-off and clouds-off LW experiments (see Figure 2).

The reduced temperature drift is likely to be valuable to studies examining the high latitudes. For example, Figure 5 (top row) shows that ACRE increases the amount of snowfall in the Arctic (total precipitation also increases; not shown). This increase in snowfall can be found whether comparing the Control to the Clouds-off LW or the Clouds-off ATM experiments, suggesting

it is a robust effect. Despite the increase in snowfall from ACRE, the surface LWCRE can have a large and offsetting role on surface snow amounts (measured as the snow water equivalent, SWE). Figure 5 shows that the role of surface LWCRE is to reduce SWE such that SWE is smaller in magnitude in the Control than in the Clouds-off LW experiment, despite the increase in snowfall. When surface LWCRE is left on in the Clouds-off ATM experiment, we see an increase in SWE, consistent with the increase in surface snowfall (Figure 5f).

### 215 3.1.4 Surface-locking

To test whether further reductions in surface temperature differences could be achieved between a complete CRE denial experiment and the control, we perform additional code modifications to directly limit the surface temperature drift. Specifically, the surface fluxes are prescribed following the methodology of Lau et al. (2019) in a new experiment, referred to as the "surface-locking" experiment. The surface-locking experiment uses the same flag settings as the clouds-off ATM experiment (TTFF).

There is one important difference between our implementation of the prescribed surface fluxes from that of Lau et al. (2019). In E3SM, we relax the surface fluxes at every time step to the prescribed surface flux to avoid triggering a numerical instability arising in the parameterization for shallow convection, turbulence, and macrophysics (the CLUBB parameterization) when the surface flux fields are overwritten directly. Pseudocode of the overwriting process is as follows.

```
    factor = 0.5
shf    = shf  + factor * (shf_input  - shf)
```




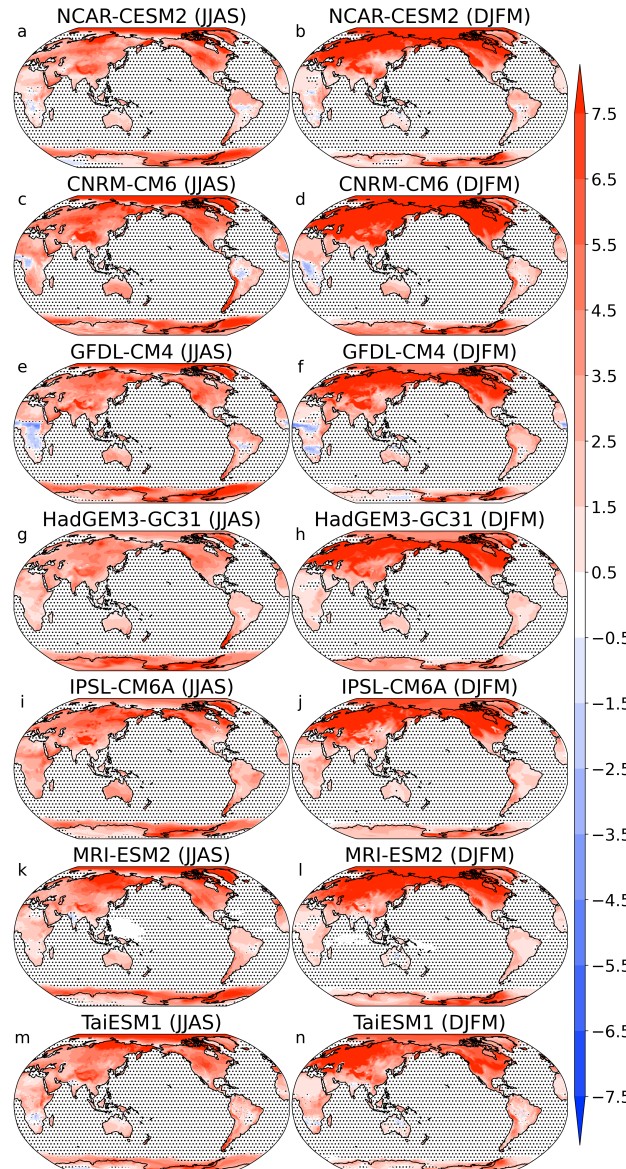

**Figure 4.** The surface temperature response to LWCRE for next generation COOKIE experiments used in CFMIP. The top row (a-g) is for JJAS and the bottom row (h-n) is for DJFM. The temperature differences are given as amip minus amip-lwoff to show the effect of clouds. Stippled areas denote regions that have no statistically distinguishable difference at the 95% confidence interval using a two-tailed Student's t-test. All values are given in K.

```
qflx   = qflx + factor * (qflx_input - qflx)
lhf    = lhf  + factor * (lhf_input  - lhf)
lwup   = lwup_input
```



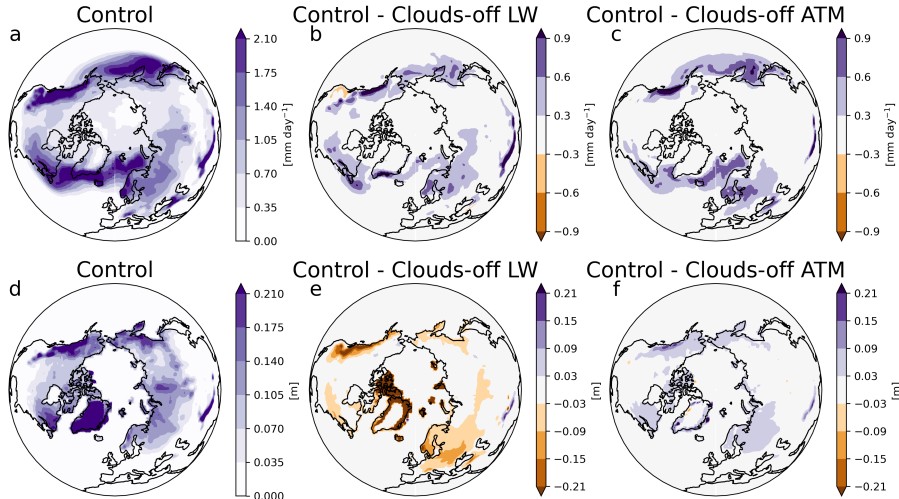

**Figure 5.** (Top row) Arctic snowfall for the (left) Control, and the impact of clouds relative to the (middle) Clouds-off LW and (right) Clouds-off ATM experiments. (Bottom row) the same, but for the snow water equivalent (SWE). Snowfall is given in $\mathrm{mm\,day}^{-1}$ and SWE is given in $\mathrm{m}$.

```
        asdir  = asdir_input
aldir  = aldir_input
        asdif  = asdif_input
        aldif  = aldif_input
```

This overwriting process is called from EAM's `phys_run2` subroutine, which occurs immediately after the coupling step and before any additional atmospheric physics parameterizations are called. The factor of $0.5$ corresponds to a nudging
timescale of 1 hour (the model has a 30 minute time step) and is ad-hoc. It was chosen empirically to minimize temperature drift while avoiding having the model crash.

In order to make use of the surface-locking functionality in E3SMv1, a multistep process is required (summarized in Figure 6). The first step is to run a control simulation to generate the surface fluxes that the experiment then makes use of. To incorporate the diurnal cycle of surface forcing, the surface flux outputs from the control simulation are stored at hourly
frequency. The model outputs needed for prescribing the surface fluxes are the sensible heat flux (`SHFLX`), moisture flux (`QFLX`), latent heat flux (`LHFLX`), surface upwelling LW radiation (`FLUS`), and surface albedo for direct visible (`ASDIR`), direct near-infrared (`ALDIR`), diffuse visible (`ASDIF`), and diffuse near-infrared (`ALDIF`). An example of specifying these fields is provided in step 1 of Figure 6. These outputs are written by the model at hourly frequency.

The second step is a processing algorithm to ready the file output for ingest into the experimental simulation. An example
script for processing these data is provided in the supplementary material (template_psld.YYYYMMDD.control.RESOLUTION.MACHINE.sh).

In addition to some minor formatting changes, the function of the processing is to composite the surface forcing onto day of the





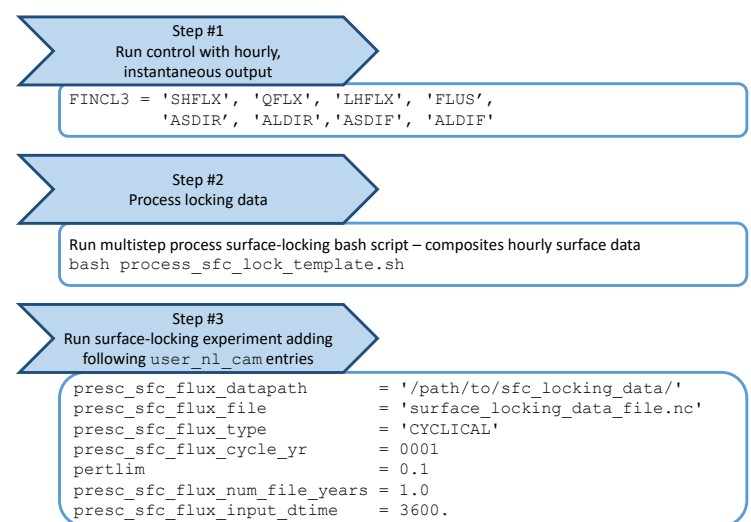

**Figure 6.** Workflow for creating the surface-locking experiment.

year and hour of the day (8760 distinct times). The compositing process retains the diurnal and seasonal cycles of the control simulation, while smoothing out the data temporally. This choice was made together with the nudging implementation described above to avoid triggering numerical instabilities. The final step in the process for running a surface-locking experiment
is to specify the appropriate namelist settings in `user_nl_cam` for the prescribed surface fluxes to be used.

The namelist settings in step 3 in Figure 6 set the path to the input file (`presc_sfc_flux_datapath`), the name of the input file (`presc_sfc_flux_file`), how to handle time points beyond the bounds of the input file time (`presc_sfc_flux_type`), the number of years in the input file (`presc_sfc_flux_num_file_years`), and the offset time needed to align the input file time with the model time (`presc_sfc_flux_input_dtime`). At the time of writing, `'CYCLICAL'` is the only option
for `presc_sfc_flux_type`.

EAMv1 solves the primitive equations using a continuous Galerkin spectral finite element method on a cubed-sphere grid (Dennis et al., 2012). The model solution is evaluated on Gauss-Lobatto-Legendre quadrature points (an example is shown in Figure 3 of Dennis et al., 2012). The uneven distribution of model columns makes it desirable to have input fields read in directly to the model grid without any interpolation. As such, code was added to facilitate reading in the prescribed surface flux
fields so that the model output from the control experiment is used without interpolation for the surface-locking experiment. This capability for reading in input fields on the model native grid is employed for the cloud-locking, prescribed-RadHt, and prescribed-CRE experiments described below as well. Appendix A provides a detailed description of this file reading capability.



Figure 2 (panels e and j) show the temperature differences between the control and surface-locking experiments. Comparing these temperature differences to those of the Clouds-off ATM suggests that there is little value to be gained from the surface-locking experiment. The increase in effort (outputting additional high frequency data from the control, processing it, then reading it into a new simulation) results in only minor reductions in temperature differences. The Northern Hemisphere mid-latitude land area (roughly 30-50°N) is one of the only areas that shows statistically significant differences between the clouds-off ATM and surface-locking experiments. Over this area, there is a halving of the temperature difference (compared to the control) going from clouds-off ATM to surface-locking during JJAS, but the differences are exacerbated at these same latitudes during DJFM (comparing Figure 2 panels d and e, as well as i and j). Qualitatively the difference patterns of these two experiments show the same response when compared to the control, and both are significantly reduced relative to the temperature differences found in the clouds-off and clouds-off LW experiments, suggesting that the clouds-off ATM is sufficient on its own for constraining the surface temperature drift and analyzing the impact of ACRE on the climate. While we continue to show results from the surface-locking experiment for completeness in this manuscript, we will not discuss its results in detail.

## 3.2 Decorrelating cloud radiative effect and circulation

### 3.2.1 Cloud-locking

The cloud-locking experiments make use of the methodology employed by Middlemas et al. (2019) to perform cloud-locking experiments in CESM1.2. The cloud-locking methodology prescribes the cloud optical properties from a control simulation into a new simulation. The variables needed for cloud locking are effective ice particle diameter (DEI), effective snow particle diameter (DES), ice gamma parameter for optics (MU), slope of droplet distribution for optics (LAMBDAC), in-cloud ice water path (ICIWP), in-cloud liquid water path (ICLWP), in-cloud snow water path (ICSWP), fraction of cloud liquid drops plus snow (CLDFSNOW), cloud fraction (CLD), and convective cloud fraction (CONCLD). These outputs are written by the model hourly to match the frequency of radiative transfer calls made by the model. To avoid any potential issues related to interpolation of the cloud fields, these variables are read in on the native grid used for the cloud-locking experiments (like the prescribed surface fluxes). Once the cloud fields have been read in, they are stored in the physics buffer (a staging area within the model for preserving data across modules or time steps). When the model computes cloud optical properties, it also stores those in the physics buffer, though under a different name. When the radiative transfer calculation needs the cloud optical properties (in the cloud_rad_props.F90 module), the flag has_prescribed_cloud (which evaluates as true when a cloud-locking input file is specified) determines whether to take the online calculated optical properties or those read from the file. Both the values calculated online and those read in from a file are stored in the physics buffer, so the logic operates by setting which index of the physics buffer the cloud radiative property calculations use (shown below).

```
if (has_prescribed_cloud) then
    i_dei    = pbuf_get_index('DEI_rad',errcode=err)
    i_mu     = pbuf_get_index('MU_rad',errcode=err)
    i_lambda = pbuf_get_index('LAMBDAC_rad',errcode=err)
```



```
        i_iciwp   = pbuf_get_index('ICIWP_rad',errcode=err)
        i_iclwp   = pbuf_get_index('ICLWP_rad',errcode=err)
        i_des     = pbuf_get_index('DES_rad',errcode=err)
        i_icswp   = pbuf_get_index('ICSWP_rad',errcode=err)
else
        i_dei     = pbuf_get_index('DEI',errcode=err)
        i_mu      = pbuf_get_index('MU',errcode=err)
        i_lambda  = pbuf_get_index('LAMBDAC',errcode=err)
        i_iciwp   = pbuf_get_index('ICIWP',errcode=err)
i_iclwp   = pbuf_get_index('ICLWP',errcode=err)
        i_des     = pbuf_get_index('DES',errcode=err)
        i_icswp   = pbuf_get_index('ICSWP',errcode=err)
    endif
```

In the above code snapshot, "_rad" denotes terms that have been read in from file and stored in the physics buffer, while those
terms lacking this "_rad" are those computed online. Currently, the cloud radiative properties are computed online regardless
of whether those values are used. While this may not be the most computationally efficient, it reduces the need for additional
code development and reduces the chances of introducing bugs.

Appendix B provides a set of step-by-step directions for generating and running cloud-locking experiments with E3SMv1
on the NERSC HPC system, but are generalizable to any system that can run E3SM. The template scripts used in this example
are included in the supplementary materials for this manuscript.

The cloud-locking experiment is fundamentally different from the complete CRE denial experiments in that it does not re-
move the mean CRE heating in the atmosphere. The cloud-locking does, however, decouple the CRE from circulation patterns,
removing any covariance between the two terms and negating the influence of CREs on the short-term evolution of atmo-
spheric motion (and vice versa). Voigt and Albern (2019) showed that the COOKIE-style (the complete CRE denial) method
and cloud-locking method offer different insight into the impact of CREs on the climate. They found the COOKIE-style method
is generally suited for understanding the role of clouds on the present-day climate, while the cloud-locking method is better
suited for understanding cloud feedbacks. Exceptions exist, of course. For example, Grise et al. (2019) used cloud-locking
experiments to quantify the impact of cloud radiative heating within extra-tropical cyclones in the current climate.

In order to compute the role of clouds on differences resulting from climate changes, cloud-locking experiments typically
include running factorial experiments where SSTs and prescribed cloud properties are toggled for multiple climate states. For
example, if '0' denotes the present-day climate, '1' denotes the +4K warming climate, 'T' denotes the SST choice, and 'C'
denotes the cloud property choice, then the four experiments would be T0C0, T0C1, T1C0, and T1C1. The cloud response
would then be

$$\text{Cloud response} = \frac{1}{2}\left((\text{T0C1} - \text{T0C0}) + (\text{T1C1} - \text{T1C0})\right) \tag{1}$$



As noted by Voigt and Albern (2019), this method can also include locking water vapor (see Voigt and Shaw, 2015). If water vapor is added to the factorial experiment design, computing all of the terms requires combining eight unique simulations (see equation 1 of Voigt and Albern, 2019). If it is assumed that water vapor must be consistent with SSTs to give credible simulations (either by locking water vapor to corresponding SSTs or by allowing for free-running water vapor), then the number of simulations required to compute the cloud response can be reduced to four. Voigt and Albern (2019) show (their Figure 1) that most features of the climate system are reliably reproduced regardless of the choices surrounding water vapor. As a result, we opt to allow for free-running water vapor in the cloud-locking simulations performed for this study, and make use of only four experiments to determine cloud responses using cloud-locking (as in equation 1).

One concern with the cloud-locking experiments comes from the data management aspect. For the standard resolution EAMv1 simulations run for this study, the data needed for cloud-locking is roughly 1.1 Tb per simulation year. As a result, only three years of cloud optical data are used for the cloud-locking experiments, with the model cycling over the input data for additional simulation years. Prior studies have found even a single year to be useful for cloud-locking (e.g., Middlemas et al., 2019), so we do not anticipate any problems related to under sampling by using three years.

We expect the cloud-locking experiment to be minimally disruptive to the circulation patterns of the atmosphere (the prescribed-RadHt and prescribed-CRE experiments described in the following sub-sections are expected to behave similarly to the cloud-locking). As a simple test of this expectation, we examine the zonal mean mass streamfunction, $\Psi$. $\Psi$ is computed as

$$\Psi(\phi, p) = \frac{2\pi a \cos\phi}{g} \int\limits_{p}^{p_{\text{sfc}}} [v] \, \mathrm{d}p \tag{2}$$

where $\phi$ is latitude, $p$ is pressure, $a$ is the Earth's radius, $g$ is the gravitational acceleration (treated as a constant in E3SM), $p_{\text{sfc}}$ is the surface pressure, and $[v]$ is the zonal mean meridional velocity. Figure 8 shows that the mean cloud radiative heating acts to amplify the circulation strength, consistent with prior studies (e.g., Harrop and Hartmann, 2016). The clouds-off experiment has a different impact than the clouds-off LW, clouds-off ATM, and surface-locking experiments, further demonstrating the complicating role of removing surface cloud radiative effects over land. The cloud-locking experiment (Figure 8 panels e and k) shows very little disruption to the seasonal mean mass streamfunction, as desired. By keeping the mean cloud radiative heating pattern, the mean circulation is largely maintained.

### 3.2.2 Prescribed-RadHt

Next, we explore two alternative approaches to the cloud-locking method which require significantly less data storage while still decoupling CREs from atmospheric circulations. The first method, described in this section, was proposed by Zhang et al. (2021b, a, 2023): instead of prescribing the cloud optical properties, the model-computed radiative heating (longwave and shortwave) at each model time step is prescribed. Unlike the cloud-locking experiments described in Section 3.2.1, the radiative effect of water vapor, as well as temperature perturbations, aerosols, and other radiatively active gases, are overwritten with their climatological values in the mean radiation experiments because mean radiation is prescribed in its entirety (not simply



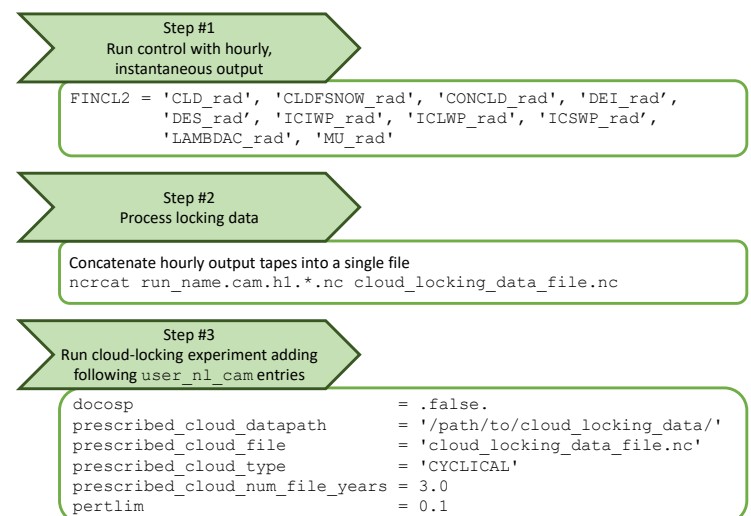

**Figure 7.** Workflow for creating the cloud-locking experiment.

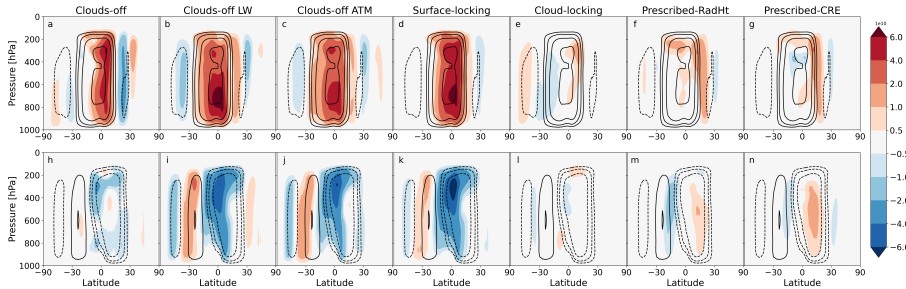

**Figure 8.** Mass streamfunction ($\Psi$) for the control experiment (contours) and the difference between the control and cloud-radiation denial experiments (colored contours). Positive values denote circulation counter-clockwise circulations (northward flow near the surface). The top row is for JJAS and the bottom for DJFM.

the ACRE). Since the climatological mean radiative heating varies smoothly from day-to-day, the input radiative heating can be prescribed using climatological values taken from monthly mean output. The monthly climatological values are linearly interpolated to the current model time at each time step. The surface fluxes are taken from the online radiative transfer cal-
culations; only the atmospheric heating rates are prescribed. For these experiments the variables needed are the LW radiative heating rate (`QRL`), the SW radiative heating rate (`QRS`), the top-of-atmosphere (TOA) net LW flux (`FLNT`), the TOA net SW flux (`FSNT`), the surface net LW flux (`FLNS`), and the surface net SW flux (`FSNS`). Note that the TOA and surface fluxes are





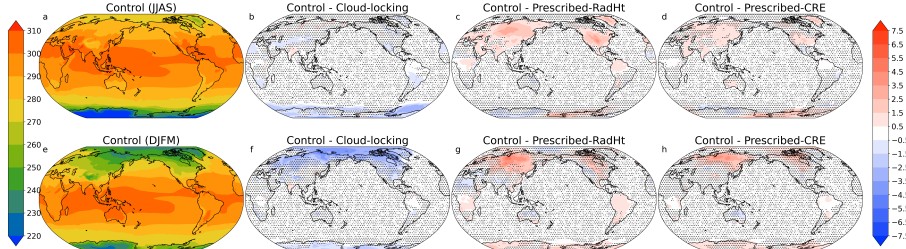

**Figure 9.** As in Figure 2, only for the decorrelation experiments: cloud-locking, prescribed-RadHt, and prescribed-CRE.

only used to enforce energy conservation since the LW and SW radiative heating profiles are output with units of $\mathrm{K\,s^{-1}}$. Our method for prescribing these heating rates, as it is currently implemented, does not resolve the diurnal cycle though future
model developments could explore that impact.

The radiative heating rate variables are fully resolved in the model vertical dimension and are provided with units of $\mathrm{K\,s^{-1}}$. To conserve energy across changing atmospheric column mass (surface pressures at any given instant will not generally equal the monthly mean surface pressure), the radiative heating rates are scaled by the ratio of prescribed net SW and LW fluxes (FSNT − FSNS and FLNS − FLNT, respectively) to the column-integrated radiative heating using the model's instanta-
neous atmospheric mass (for a hydrostatic model like EAM, this is defined as the pressure thickness divided by the gravitational acceleration). The algorithm needed to accomplish this scaling is provided below (and is implemented as subroutine conserve_radiant_energy in the prescribed_radheat.F90 module).

$$QRS^k_{scaled} = QRS^k_{input}\left(\frac{FSNT - FSNS}{\sum_k QRS^k_{input}\delta^k p/g}\right) \tag{3}$$

$$QRL^k_{scaled} = QRL^k_{input}\left(\frac{FLNS - FLNT}{\sum_k QRL^k_{input}\delta^k p/g}\right) \tag{4}$$

where $\delta^k p$ is the pressure thickness of model level $k$ and $g$ is the gravitational acceleration (assumed constant in EAM).

The radiative heating is only prescribed within the troposphere. Rather than use the diagnosed tropopause for each atmospheric column, we use a static mask for each column based. A fixed set of coefficients is used to determine which levels use the prescribed radiative heating, which use the online computed radiative heating, and which are a blend of both. The transition zone, where the radiative heating is a weighted combination of prescribed and online computed heating occurs roughly around
pressure levels 25-80 hPa. The code to accomplish this is done in EAM's radheat_tend subroutine within radheat.F90.

```
qrs(i,k) = (1._r8 - p_radht_coefs(k)) * qrs(i,k) &
          + p_radht_coefs(k) * cpair * qrs_input(i,k)
qrl(i,k) = (1._r8 - p_radht_coefs(k)) * qrl(i,k) &
          + p_radht_coefs(k) * cpair * qrl_input(i,k)
```





The `i` and `k` coefficients denote atmospheric column and level, respectively. For reference, the coefficients, `p_radht_coefs` are defined below.

```
p_radht_coefs(1:25) = (/ &
     0.000_r8, 0.000_r8, 0.000_r8, 0.000_r8, 0.000_r8, &
     0.000_r8, 0.000_r8, 0.000_r8, 0.000_r8, 0.000_r8, &
     0.000_r8, 0.000_r8, 0.000_r8, 0.000_r8, 0.000_r8, &
     0.125_r8, 0.250_r8, 0.375_r8, 0.500_r8, 0.625_r8, &
     0.750_r8, 0.875_r8, 1.000_r8, 1.000_r8, 1.000_r8 /)
p_radht_coefs(26:pver) = 1.000_r8
```

We hypothesize that the prescribed mean radiation experiment will show similar behavior to the cloud-locking experiments.

To test that hypothesis we run a similar factorial set of experiments where SSTs for present day and with +4K warming are each combined with both mean heating from a control simulation with present-day SSTs and one from a control simulation using +4K warming. The clear-sky radiative heating depends on the SSTs used, so for the T0C1 and T1C0 experiments the prescribed mean radiative heating is a combination of clear-sky heating consistent with the SSTs and cloud radiative heating consistent with the cloud fields. For example, the prescribed SW radiative heating for the T0C1 experiment is constructed as

follows.

$$\mathrm{QRS_{T0C1}} = \mathrm{QRS_{clr,0}} + \left(\mathrm{QRS_{all,1}} - \mathrm{QRS_{clr,1}}\right) \tag{5}$$

where subscript 0 and 1 refer to values from a free-running control simulation with present-day and +4K warming SSTs, respectively. The same equation is used to compute the T0C1 values of FSNT, FLNT, FSNS, FLNS, and QRL.

Like cloud-locking, the prescribed mean radiation has little impact on the surface temperature or circulation (Figures 9 and

8). Further comparison is provided in section 4.2.

### 3.2.3    Prescribed-CRE

The second alternative to cloud-locking simply prescribes the CRE, following similar methodology as the prescribed-RadHt experiment detailed above in section 3.2.2. By prescribing CRE directly, we allow clear-sky radiative heating from water vapor to match the distribution of water vapor in the atmosphere, similar to letting water vapor freely evolve in the cloud-locking

experiments. This prescribed CRE methodology also negates the need of creating cross-experiment input files taking the clear-sky radiative heating from one climatology and the CRE from another. Instead, the user only needs to select the appropriate CRE input file. The CRE data read in from file is combined with the clear-sky radiative heating in the `radiation_tend` submodule. Pseudocode of this process is shown below.

```
qrs(i,k) = (cpair * qrs_cld(i,k)) + qrs_clr(i,k)
```
```
qrl(i,k) = (cpair * qrl_cld(i,k)) + qrl_clr(i,k)
fsnt(i)  = fsnt_cld(i,1) + fsnt_clr(i)
```



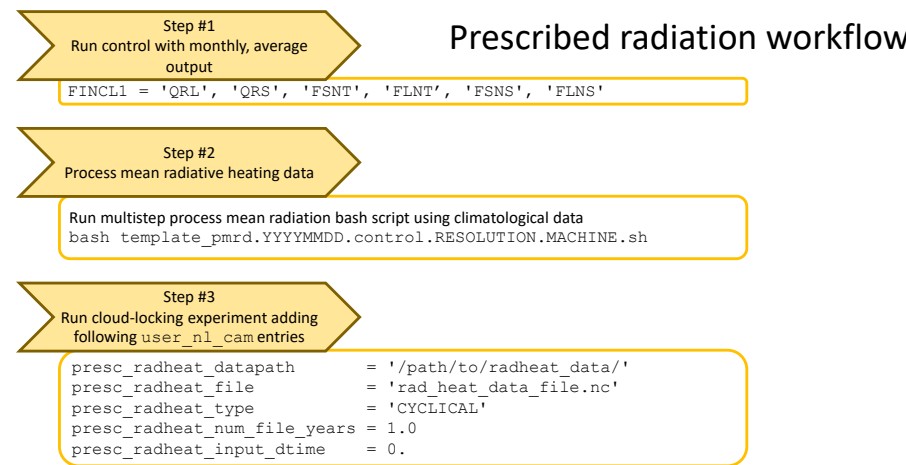

**Figure 10.** Workflow for creating the prescribed-RadHt experiment.

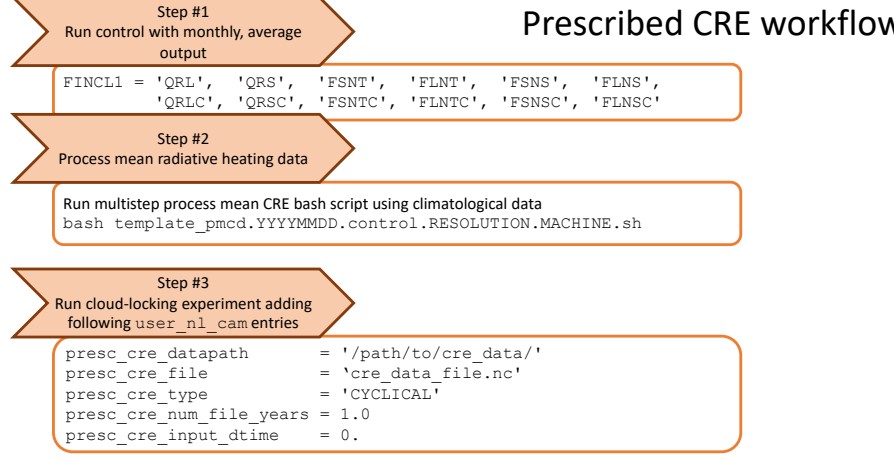

**Figure 11.** Workflow for creating the prescribed-CRE experiment.

```
flnt(i)  = flnt_cld(i,1) + flnt_clr(i)

fsns(i)  = fsns_cld(i,1) + fsns_clr(i)

flns(i)  = flns_cld(i,1) + flns_clr(i)
```

Note that `cpair`, the specific heat of dry air at constant pressure, is already included in `qrs_clr` and `qrl_clr`, so multiplying `qrs_cld` and `qrl_cld` by `cpair` makes the units consistent across terms.



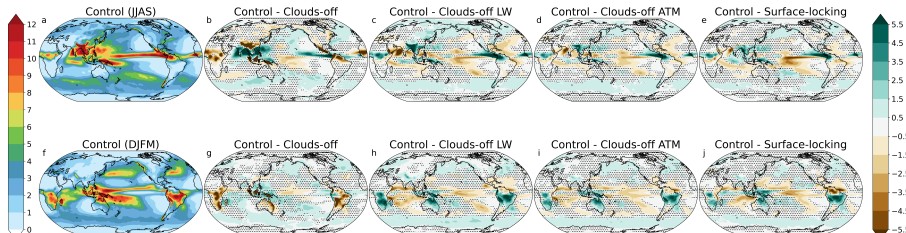

**Figure 12.** As in Figure 2, but for precipitation. All values have units of $\mathrm{mm\,day}^{-1}$.

## 4 Results

To validate our suite of simulations, we perform several analyses aimed to examine the impacts of CREs in E3SM and place them in context of prior work. We examine the intensity of monsoon rainfall to the mean CRE, the impact of cloud feedback on several circulation metrics, and finally the response of the precipitation distribution to CREs.

### 4.1 Monsoon rainfall

In this section, we are interested in answering the question, "How do cloud radiative processes affect the seasonal mean structure of tropical precipitation?" While there are many facets to the ITCZ structure, for this analysis, we focus on the intensity of the water cycle as measured by surface precipitation minus evaporation. Much of the preceding work outlined in the introduction section has been done using zonally symmetric aquaplanets, which makes for simple measures that characterize the zonal mean ITCZ well (e.g., Popp and Silvers, 2017; Fläschner et al., 2018). When using realistic land-sea geography, however, one must account for the possibility of different responses over different regions. Figure 12 shows that ACRE generally increases precipitation over the ascending portions of the Hadley/Walker circulation, consistent with earlier findings (Slingo and Slingo, 1988; Sherwood et al., 1994). To quantify this change in the tropical water cycle, we make use of the Normalized Gross Moist Stability (NGMS) framework of Harrop et al. (2018, 2019). NGMS is defined as the ratio of moist static energy (MSE) export to import of moisture.

$$\Gamma = -\frac{\nabla \cdot \{\mathbf{v}h\}}{L\nabla \cdot \{\mathbf{v}q\}} \tag{6}$$

where $\Gamma$ is the NGMS, $\mathbf{v}$ is the horizontal wind vector, $h$ is the MSE, and $L$ is the latent heat of vaporization (considered a constant in EAMv1). This NGMS framework provides an energetic perspective for diagnosing and attributing changes in the water cycle ($P$-$E$). We use a simplified version of equation 5 from Harrop et al. (2019):

$$\Delta(P-E) = \frac{\Delta\mathrm{ACRE}}{L\Gamma} + \frac{\Delta\mathrm{THFLX}}{L\Gamma} + \nabla \cdot \{\mathbf{v}q\}\frac{\Delta\Gamma}{\Gamma} + \mathrm{Residual} \tag{7}$$

where $E$ is the surface evapotranspiration, THFLX is the total surface turbulent heat flux (sensible plus latent), and the residual term groups other factors such as the net clear-sky radiative heating within a column, storage terms, and non-linear interactions.





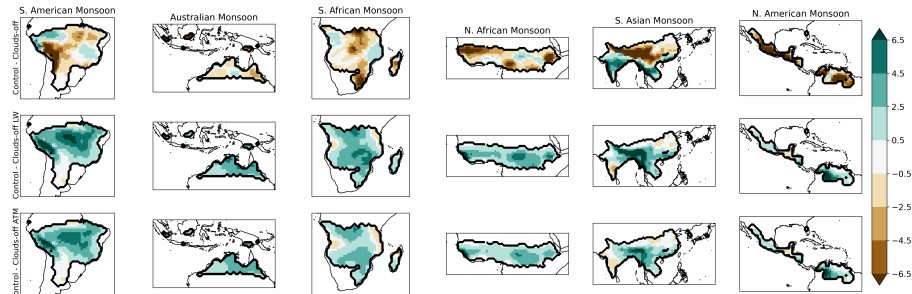

**Figure 13.** Differences in $P-E$ for the six subtropical land monsoon regions (differences are control minus experiment). The top row is for the clouds-off experiment, the middle row is for the clouds-off LW experiment, and the bottom row is for the clouds-off ATM experiment. The border of each monsoon region is outlined in a thick black contour. All values have units of $\mathrm{mm\,day^{-1}}$.

The first two terms on the right hand side provide an estimate of the rainfall change associated with changes in ACRE and surface turbulent heat fluxes, respectively. A simple conceptual understanding of NGMS is that it is an inverted measure of how much precipitation is produced per unit of energy export by the circulation (assuming all of the moisture convergence falls out as precipitation). If that ratio does not change (NGMS constant), and there is more heating in the column, then the circulation intensifies to export the additional energy and restore balance, which also increases precipitation. As such, this diagnostic framework links the tropical hydrological cycle to energy perturbations, and we expect ACRE heating within the column to result in an increase in surface $P-E$.

We focus on the subtropical land monsoon areas for our intensity analysis. To separate the intensity changes from the area changes, we use a static mask to define the monsoon region. This mask uses the Global Precipitation Climatology Project (GPCP) one-degree daily (1DD) data (Huffman et al., 2001, 2009) and the monsoon criterion of Wang and Ding (2008). The bounds of these masks can be seen in Figure 13 along with the change in precipitation minus evapotranspiration between the control and the clouds-off (top row), clouds-off LW (middle row), and clouds-off ATM (bottom row) experiments in color shading.

The NGMS terms are provided in Figure 14 for each of the monsoon regions (rows) and for the cloud response as measured by comparing the control with the clouds-off (left column), clouds-off LW (middle column), and clouds-off ATM (right column) experiments. There are several interesting results from Figure 14 that can be discerned by comparing across monsoon regions and different experiment types.

First, ACRE always acts to increase precipitation in the monsoons. The response of precipitation to ACRE is not surprising given that ACRE provides additional energy to the atmospheric column that, all else equal, would require a stronger circulation to export that energy, drawing in more moisture to fuel precipitation. The amount of precipitation increase attributed to ACRE varies depending on the mean ACRE of the monsoon region, but is similar across experiments (comparing green bars across experiments). There is a slight increase in the ACRE term when atmospheric SWCRE is included in addition to atmosphere LWCRE, as is the case for the clouds-off ATM and clouds-off experiments.





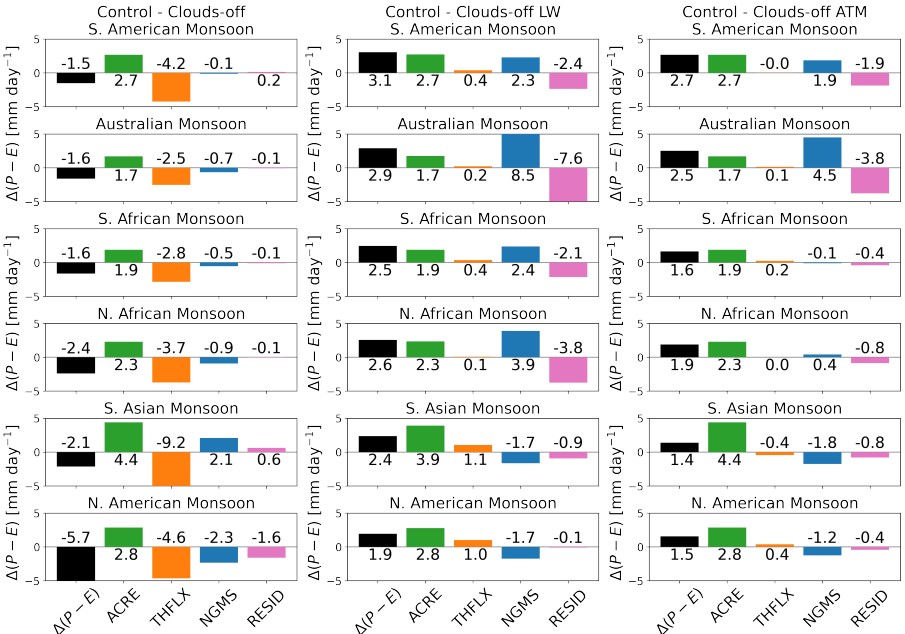

**Figure 14.** Differences in $P-E$ for the six subtropical land monsoon regions (differences are control minus experiment) and those differences broken down into terms (see equation 7). The left column is for the clouds-off experiment, the middle column is for the clouds-off LW experiment, and the right column is for the clouds-off ATM experiment. The rows represent the different monsoon regions, from top to bottom: the South American, Australian, South African, North African, South Asian, and North American monsoons. All values have units of $\mathrm{mm\,day}^{-1}$.

Second, the surface flux response to changing surface CRE diverges depending on whether surface SW or LW CREs dominate the response. In the middle column, where only surface LWCRE changes between the control and experiment (clouds-off LW), the THFLX is positive across all monsoon regions, meaning the role of surface LWCRE is to increase THFLX and

increase precipitation. In the left column, however, where surface SWCRE also changes between the control and experiment (clouds-off), the THFLX term is negative across all monsoon regions. Again, this is consistent with the expectation that surface SWCRE reduces the THFLX and, hence, reduces the precipitation (see discussion in section 3.1.1). In the right column, where the surface "sees" the CREs in both the control and experiment (clouds-off ATM), the precipitation responses oscillate around $0\,\mathrm{mm\,day}^{-1}$, with no consistent response across monsoon regions.

Third, there is no consistent response in the NGMS term across monsoon regions. As noted above, it is expected that the circulation increases in intensity over the land monsoons as a result of ACRE, and the vertical velocity at 500 hPa does increase in magnitude over these regions (not shown). The relation between MSE export and moisture import, however, does not behave uniformly in response to that circulation change. This result suggests the dynamical responses to ACRE depend on the geography of the monsoon region. The South Asian and North American monsoons have a different response compared to

the other monsoons, and they also have the most complicated orography, which may play an important role. Where the NGMS





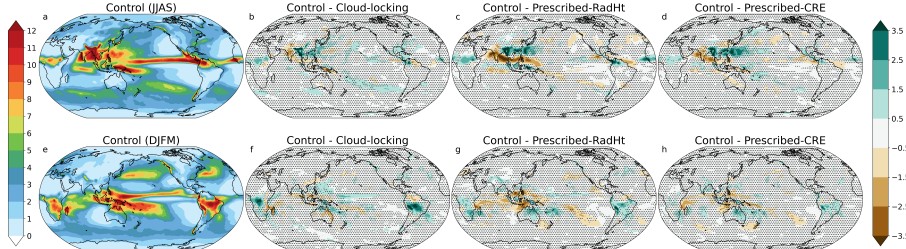

**Figure 15.** As in Figure 12, only for the decorrelating experiments: cloud-locking, prescribed-RadHt, and prescribed-CRE. All panels have units of $\mathrm{mm\,day}^{-1}$.

term is largest, there also tends to be a strong cancellation from the residual term, coming primarily from the non-linear term (not shown), further suggesting complicated dynamic responses to ACRE for the monsoon regions. A deeper analyses of these dynamical responses is beyond the scope of the current manuscript.

In short, the NGMS framework correctly identifies the expected responses when comparing across experiments for the

various terms. ACRE increases monsoon precipitation, but the decrease in precipitation owing to surface SWCRE can mask the role of ACRE in experiments like clouds-off. For the monsoons, the role of surface LWCRE is small, but positive, such that the increase in $P - E$ owing to CREs is larger when comparing with clouds-off LW than with clouds-off ATM across the various monsoon regions.

The above results are consistent with those from Byrne and Zanna (2020), despite their use of axisymmetric aqua planets.

Their setup includes a slab-ocean configuration with a shallow mixed layer depth (5 m) and no horizontal heat transfer, allowing for it to have a strong seasonal temperature signal akin to real world monsoon systems. The use of this slab-ocean also allowed for Byrne and Zanna (2020) to examine both the SW and LW CREs separately, and they found the SWCRE dampens monsoon intensity while LWCRE amplifies monsoon intensity. Our results show their findings hold in a more realistic modeling setup.

Unsurprisingly, the precipitation response of the CRE-circulation interactions (as measured by the cloud-locking, prescribed-

RadHt, and prescribed-CRE experiments) differs from the COOKIE-style experiments, though their precipitation response is similar qualitatively across the decorrelation experiments and between seasons. It is interesting to note that the Boreal summer precipitation response resembles the EOF pattern of the Boreal Summer Intraseasonal Oscillation (BSISO) (compare to Figure 2 of Kikuchi, 2021). The moisture mode theory that is a key component of BSISO (Kikuchi, 2021) is modulated by ACRE (Adames and Kim, 2016), so it is perhaps not too surprising that these would be connected. As the leading mode of variability

for the summer in this region (Kikuchi, 2021), forcing changes are likely to prompt responses that excite this mode. We see that the seasonal mean response resembles the EOF pattern of BSISO. The response seen in Figure 15 suggests that the covariance of CREs and circulation may alter the distribution of phases of BSISO. In theory, such a distribution change could occur from a change in the BSISO longevity relationship according to phase shown in Figure 10 of Kikuchi (2021). For example, one might expect a dampening of the progression of BSISO without the covariance between CRE and circulation. Future research




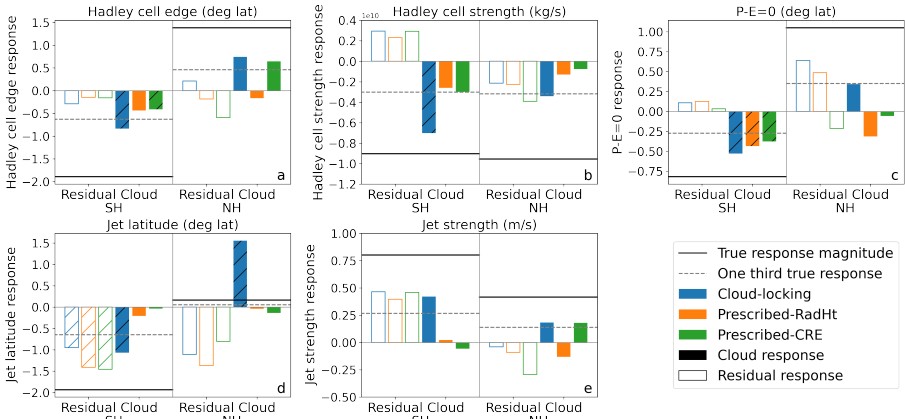

**Figure 16.** Responses to warming (+4K SSTs minus present-day) for Hadley cell edge (a), Hadley cell strength (b), poleward edge of the subtropical dry zone (c), extratropical eddy-driven jet latitude (d), and the strength of the eddy-driven jet (e). The true response value (as measured using the control and control +4K experiments) is shown in the black solid line, and one third of that value is shown in the gray, dashed line of each panel. The unfilled bars are the residual term and the filled bars are the cloud term. Hatching indicates results where the difference in means are statistically different from zero at the 95th percentile. Since the circulation is not perfectly symmetric across the equator, both Southern and Northern Hemisphere values are provided in each panel. The blue bars are for the cloud-locking experiment; the orange bars are the prescribed-RadHt experiment; and the green bars are the prescribed-CRE experiment. Units for each panel are provided in the panel title.

is needed to test these hypotheses and better understand the role of CRE-circulation covariations on the water cycle and its intraseasonal modes of variability.

### 4.2    Cloud feedback influence on circulation metrics

Next, as noted in section 3.2.1, we use the same decomposition as Voigt and Albern (2019, their equation 5) to examine the impacts of removing the cloud circulation contributions to the general circulation. For each of the cloud-locking, prescribed-

RadHt, and prescribed-CRE experiments, we compute the same circulation metrics used by Voigt and Albern (2019): the Hadley cell edge; the Hadley cell strength; the poleward edge of the subtropical dry zone; the extratropical eddy-driven jet latitude; and the strength of the eddy driven jet. The definitions of these metrics are the same as those used by Voigt and Albern (2019), which we reproduce here for convenience. The Hadley cell edge is the latitude at which the zonal mean stream function associated with the Hadley cell (between 30° N/S) goes to zero at 500 hPa. The Hadley cell strength is the maximum of the

zonal mean stream function between 200 hPa and 850 hPa. The subtropical dry zone edge is the latitude near 40° N/S where $P - E = 0$. The jet latitude is the location of the maximum 850 hPa zonal wind, and its strength is the value of that maximum. Like Voigt and Albern (2019), we follow Barnes and Polvani (2013) and fit a quadratic function around the location of the maximum wind on a 0.01° grid to find the jet latitude and strength.





Figure 16 shows each of these metrics for the annual mean of each hemisphere. Voigt and Albern (2019) note that assessing
the cloud impact is only relevant when the residual term is less than one third of the total change (as measured using the control
+4K and present-day climatology runs). The total value is marked as a solid black horizontal line, and one third of its value
is provided as a gray, dashed line. The residual terms for each of the cloud-locking, prescribed-RadHt, and prescribed-CRE
experiments are given in the non-shaded bars, and their cloud contribution terms are provided in the shaded bars. The residual
is computed as in Voigt and Albern (2019, their equation 3), which is the total response minus the sum of the Cloud response
(equation 1) and SST response (akin to equation 1, only with T1-T0 terms).

Ideally, the residual terms would all be smaller in magnitude than the gray, dashed line for each subplot. Where this is not
the case, the separation of the response into cloud and SST terms is not well-suited for that particular metric. Hatching denotes
where the metrics are statistically different from zero (at the 95th percentile using a two-tailed t-test).

The residual terms exceed the one third threshold for the jet latitude and jet strength metrics in both hemispheres across the
experiments. This suggests the cloud contribution metric computed here may not be reliable for assessing the role of clouds on
the jet, and a more careful analysis would be needed to understand the cloud feedback impact on the jet. The residual terms are
generally less than the one third threshold for the Hadley cell metrics. Except for the Hadley cell edge response in the Northern
Hemisphere, the cloud responses agree across the experiments, suggesting the cloud impact on expanding the Hadley cell edge
and decreasing the strength of the circulation may be robust, despite many of the bars failing the statistical significance test.
For the Hadley cell metrics, the contribution from the cloud response term is generally larger in the cloud-locking experiment
than the other two. One hypothesis is that the small time-scale effects that are missed by the monthly mean data used in the
prescribed-RadHt and prescribed-CRE experiments are important to the changes in the Hadley cell. This point will need to be
examined further in future work.

Finally, for the subtropical dry zone expansion, there is a robust and significant response in the Southern Hemisphere, with
the three experiment types agreeing in both sign and magnitude and the residual terms being relatively small. The change
in Southern Hemisphere subtropical dry zone edge in the cloud-locking experiment is consistent with the changes in Hadley
cell edge and jet latitude for this experiment, suggesting that cloud-locking may have an advantage over the other covariance-
denial experiments for examining circulation changes owing to cloud feedbacks. The poleward expansion of the Hadley cell
edge, subtropical dry zone, and midlatitude jet are consistent with the results of Voigt and Albern (2019). For the Northern
Hemisphere, however, the residual terms are large for both the cloud-locking and prescribed-RadHt experiments, and there
is no consistent response in the cloud term (not even the sign is consistent across experiments). We speculate that the role
of clouds is less important in the Northern Hemisphere owing to the strong zonal asymmetry and role of stationary waves in
co-evolving circulation changes (Wills et al., 2019), but this requires future research efforts to understand.

In short, these three experiments are generally in agreement in terms of whether the decomposition is reliable, and where
they are reliable, are more often than not in agreement on the sign of the cloud response. There is some indication that the
cloud-locking experiment has more robust changes, particularly for the jet response, but more research is needed to under-
stand these differences. While fully understanding the differences between experiments requires further research, the results




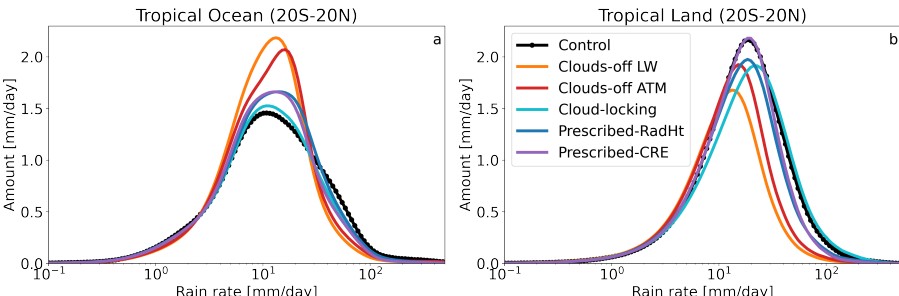

**Figure 17.** Hourly rain rate amount distributions for (a) tropical ocean regions and (b) tropical land regions. All experiments use present-day SSTs and units are given in $\mathrm{mm\,day^{-1}}$.

presented within this manuscript suggest optimism toward using the prescribed-RadHt and prescribed-CRE experiments as computationally cheaper alternatives to cloud-locking.

### 4.3 Rain rate distribution

Finally, we examine the distribution of tropical precipitation separated over ocean and land. The amount distribution is computed following Pendergrass and Hartmann (2014), with a minimum rain rate of $0.03\ \mathrm{mm\,day^{-1}}$ and a bin size growth rate of 7%. Figure 17 shows the hourly amount distribution (the amount of rainfall accumulated in each rain rate bin) for the control, clouds-off LW, clouds-off ATM, cloud-locking, prescribed-RadHt, and prescribed-CRE experiments. Figure 17a shows that the removal of CREs reduces the amount of oceanic precipitation occurring at intense rain rates (in excess of $30\ \mathrm{mm\,day^{-1}}$) while increasing the amount of rain falling at weaker rain rates. This same pattern is true even when only the covariance of CREs and the circulation are disrupted, but to a lesser degree. Our results agree with the findings of Medeiros et al. (2021), who found that removing the mean CRE or using cloud-locking reduces the occurrence rate of intense precipitation over tropical oceans.

Over land (Figure 17b), the responses are more complicated. The complete CRE denial experiments see a decrease in rain amount at high rain rates and little change at low rain rates. The three covariance denial experiments show very different behavior from one another. The cloud-locking experiment preserves the intense rain characteristics, while reducing the amount of rain falling at low rain rates. The prescribed-RadHt experiment reduces the amount of rain falling for all rain rates exceeding roughly $10\ \mathrm{mm\,day^{-1}}$, while the prescribed-CRE experiment only reduces the amount of rain falling for rain rates exceeding roughly $30\ \mathrm{mm\,day^{-1}}$. Future work is needed to better understand this land response — in particular, what spatial and temporal scales give rise to the differences.

### 5 Conclusions

In this manuscript, we document a series of experiments run with E3SMv1 meant to examine the impact of CREs on the circulation and water cycle. The variety in these experiments helps us better understand the role of CREs separated into LW





and SW components, as well as their relative impacts within the atmosphere and at the surface. These experiments can also help guide future modeling efforts. When resources limit the number of experiments that can be performed, we recommend using the clouds-off ATM experiment for studies interested in better understanding the impact of ACRE on the present-day circulation. The clouds-off ATM experiment design is also well-suited for simulations that use active ocean and sea-ice models, since there is much less impact on surface radiative fluxes and temperatures than in the clouds-off or clouds-off LW experiments.

We have also demonstrated two CRE-circulation decorrelating experiment alternatives to the cloud-locking design: the prescribed-RadHt and prescribed-CRE. While not identical, there is general agreement across these three experiment types for the responses in precipitation and the Hadley circulation. The prescribed-RadHt and prescribed-CRE experiments are of particular interest because these fields can more readily be taken from other models, reanalyses, or observations to be used to quantify the role of CRE biases on the circulation and water cycle. Future work is needed to test whether monthly data is the ideal frequency for the prescribed-RadHt and prescribed-CRE experiments or whether shorter timescale variability is needed. If monthly data can be shown to be sufficient, then the prescribed-RadHt and prescribed-CRE experiment types will continue to be sufficiently less data intensive than cloud-locking, making them an appealing alternative for high resolution modeling or for examining interannual or interdecadal variability associated with climate modes like El Nino Southern Oscillation or the Indian Ocean Dipole.

This manuscript documents the code changes that allow for these experiments to be run within E3SMv1. Namelist settings have also been provided to reproduce the experiments in future simulations. Template scripts are provided in the supplementary material to process the input data needed for the surface-locking, cloud-locking, prescribed-RadHt, and prescribed-CRE experiments. We also demonstrate several results related to the role of CREs on the monsoon circulations, several circulation metrics' response to warming, and the distribution of rain amount. These results serve as an example of the types of questions this simulation suite is well-suited to answer, and also place these results within the context of prior findings.

The output from these experiments is a valuable community resource, as is the capability of E3SM to run these types of CRE experiments. As E3SMv2 has been made available to the community after these experiments were completed, the code has been updated such that all of these experiments can be run with E3SMv2 (see the code availability statement for the repository where the code is hosted).

*Code and data availability.* The source code needed to run the experiments is available at https://github.com/beharrop/E3SM. The code to generate the figures is available at Zenodo (10.5281/zenodo.8072504). The supplementary material containing the template scripts can also be found at Zenodo (10.5281/zenodo.8125770). The simulation output from this study are made available at https://portal.nersc.gov/archive/home/b/beharrop/www/e3sm_cre_denial_overview_data/Data_Overview_CRE_denial_in_E3SM.tar. The COOKIE2 results rely on publicly available data available through the Earth System Grid Federation which can be accessed at https://esgf-node.llnl.gov/search/cmip6/. The CERES-EBAF data are also publicly available for download at https://asdc.larc.nasa.gov/project/CERES/CERES_EBAF_Edition4.1





## Appendix A: Reading native grid data

The new code used to read in the model surface fluxes, cloud optical properties, radiative heating, or CRE can be found in `prescribed_surface_flux.F90`, `prescribed_cloud.F90`, `prescribed_radheat.F90`, and `prescribed_cre.F90` respectively. The codes have only minor differences between them, so for the purpose of describing them we will describe those found in `prescribed_surface_flux.F90`. All three files also draw heavily upon the `input_data_utils.F90` module file.

A data structure called `presc_sfc_flux_type` is created to store the data. It allows for two time slices of the input files to be read in for temporal interpolation. The data are expected to have dimensions matching the current simulation. One caveat is that all data are expected to have a level dimension (dimension name 'lev'), even for data that are output without a level dimension (such as surface sensible heat flux). The addition of the level dimension is needed as a result of a bug in the `infld` routines used to read from the files. This bug is an open issue for the E3SM project, and until it is fixed, adding the level dimension to all fields is a necessary work around.

The advancement of the time coordinate is handled by subroutines contained within `input_data_utils.F90`. The key subroutines are those that assign the weights for the two time slices for interpolation. These routines handle either serial or cyclical data streams. For serial data, the model time must be within the time bounds of the input file. For cyclical data, if the model time exceeds the upper bound of the file time, then that upper bound is subtracted off of the model time until it falls within the bounds (like a modulo operator).

## Appendix B: Steps for running cloud-locking on NERSC

The experiments shown throughout this manuscript make use of E3SMv1. E3SMv2 was released during the writing of this manuscript, and the instructions for using cloud-locking have been updated to allow for either E3SMv1 or E3SMv2 to be used. All of the sample scripts are archived in `e3sm_cre_templates.tar` within the supplementary materials.

**Step 1:** clone the repository

*Disclaimer* the following steps are different for E3SMv1 and E3SMv2 owing to changes in how CIME and other submodules are incorporated into the model that were made between v1 and v2. It is also important to note that these instructions are designed to work with the current NERSC computing environment. NERSC updates its libraries and systems periodically, and changes may be required to the E3SM source code to compile the model after system maintenance and updates. While these steps should be generalizable to many high performance computing systems, it is beyond the scope of this manuscript to provide guidance for getting E3SM to run on new computing systems.

**Clone v1 repository**

```
git clone --recursive git@github.com:beharrop/E3SM.git
cd ./E3SM
```




```
git submodule update --init
git submodule deinit cime
git checkout beharrop/atm/cre_experiments
git submodule update --init
```

**Clone v2 repository**

```
git clone --recursive git@github.com:beharrop/E3SM.git
cd ./E3SM
git submodule update --init --recursive
git checkout beharrop/atm/cre_experiments_v2
git submodule update --init --recursive
```

**Step 2:** run a control simulation and generate the cloud optical property output

Modify `simple_e3sm_script.YYYYMMDD.control_v*.RESOLUTION.MACHINE.sh` to point at your code and output directories. Then run the script.

```
bash simple_e3sm_script.YYYYMMDD.control_v1.RESOLUTION.MACHINE.sh
OR
bash simple_e3sm_script.YYYYMMDD.control_v2.RESOLUTION.MACHINE.sh
```

**Step 3:** process the cloud optical property output files into a cloud-locking input dataset

Modify `template_pcld.YYYYMMDD.control.RESOLUTION.MACHINE.sh` with the name and directory of your
output as well as the number of years desired for concatenation. Then run the script.

```
sbatch template_pcld.YYYYMMDD.control.RESOLUTION.MACHINE.sh
```

**Step 4:** run a cloud-locking experiment

Modify `simple_e3sm_script.YYYYMMDD.cld_lock_v*.RESOLUTION.MACHINE.sh` to point at your code
and output directories. Then run the script.

```
bash simple_e3sm_script.YYYYMMDD.cld_lock_v1.RESOLUTION.MACHINE.sh
OR
bash simple_e3sm_script.YYYYMMDD.cld_lock_v2.RESOLUTION.MACHINE.sh
```

**Step 5:** process climatologies for analysis

Modify `process_climos_YYYYMMDD.control_and_cld_lock.sh` with the locations, names, and mapping files
(for different grids). Then run the script.



```
sbatch process_climos_YYYYMMDD.control_and_cld_lock.sh
```

*Author contributions.* BEH, JL, and LRL conceptualized the manuscript. BEH, WKML, KMK, BM, BJS, GAV, BZ, and BS contributed to methodology and software. BEH led the investigation and wrote the original draft. All authors contributed to the review and editing process.

*Competing interests.* The authors declare that they have no conflict of interest.

*Acknowledgements.* This study is supported by the U.S. Department of Energy Office of Science Biological and Environmental Research (BER) as part of the Regional and Global Model Analysis program area. The Pacific Northwest National Laboratory (PNNL) is operated for DOE by Battelle Memorial Institute under contract DE-AC05-76RLO1830. This research was performed using BER Earth System Modeling program's Compy computing cluster located at Pacific Northwest National Laboratory. PNNL is operated by Battelle for the U.S. Department of Energy under Contract DE-AC05-76RL01830. Additional analyses were performed using resources of the National Energy
Research Scientific Computing Center, a DOE Office of Science User Facility supported by the Office of Science of the U.S. Department of Energy under Contract No. DE-AC02-05CH11231. BM acknowledges support from the DOE RGMA under Award Number DE-SC0022070 and National Science Foundation (NSF) IA 1947282, and also the National Center for Atmospheric Research (NCAR), which is a major facility sponsored by the NSF under Cooperative Agreement No. 1852977. This research was performed using BER Earth System Modeling program's Compy computing cluster located at Pacific Northwest National Laboratory. PNNL is operated by Battelle for the U.S. Department
of Energy under Contract DE-AC05-76RL01830. Some of the figures presented herein were generated in part using E3SM Diags (Zhang et al., 2022a, b). NCO (Zender, 2008; Zender et al., 2022) was used to generate climatologies and for data regridding.



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
