# Peer review of "An overview of cloud-radiation denial experiments for the Energy Exascale Earth System Model version 1"

_EGUsphere, 2023_

## Author Response (AR2)

**Reviewer #1**

The manuscript of Harrop et al., 2023 has two goals:

to document changes in E3SM model needed to perform several cloud-radiation denial experiments and provide instructions on how to run them

describe and interpret results of these experiments with the aim of giving guidance on what experiments needed.

While the utility of (1) is limited to E3SM model users, (2) provides several useful insights on cloud-radiation denial experiments with a broader relevance and is therefore appropriate for GMD. The manuscript is well structured, provides several interesting findings, and can be accepted after the comments are addressed.

We thank the reviewer and the team for their insightful comments and suggestions for improvement.

General comments:

It would be great if the authors could provide a more systematic overview of what prescribing monthly mean radiative heating/CRE (Prescribed-Ht and Prescribed-CRE) methods can and what cannot do compared with Cloud-locking.

We can't hope to create an exhaustive list with these experiments as they are relatively new. We hypothesize that the different timescales of the prescribed cloud optical properties (hourly), radiative heating (monthly climatology), and CRE (monthly climatology) may be important for understanding their differences. The impact of combining the water vapor radiative effect with the CRE in the prescribed radiative heating experiment, may also be an important factor for the differences between it and the prescribed-CRE experiments. Voigt and Albern (2019) found that locking water vapor versus free-running water vapor simulations produce the same qualitative results, but small quantitative differences can arise, like we see between our prescribed radiative heating and prescribed CRE experiments. We hope that we and the community can continue to make use of these experiments and learn more about their differences.

In particular, it would be interesting to get more information about convection in the Prescribed-Ht and Prescribed-CRE experiments. What causes the changes in the simulated rainfall distribution over tropical land? Why does the frequency of high rain rates decrease? How does this compare to cloud locking?

Better understanding the convective changes that give rise to the differences in rain rate amount distributions between the different experiments is of interest to us as well. Medeiros et al. (2021) suggest that it is not the large-scale environment, but the way convection organizes that changes the extreme rainfall. A simple check of the thermodynamic profile changes (control minus experiment) shows this is the case for our E3SM experiments as well (see below). Even though the oceanic rain rate amount shows similar behavior across all the experiments, the average change in temperature and

moisture does not even agree on sign for the oceanic regions. Likewise, over land, similarity in the large-scale environmental changes does not correspond to similarity in rain rate amount differences. Examining the convective organization in these experiments is a more involved process and requires a level of analysis and care better suited for follow up work. We have added the following lines to the revised manuscript in section 4.3, "Medeiros et al. (2021) found that it was not the large-scale environment, but the way convection organized in the simulations that was important for the changes in extreme precipitation related to CREs. There is no consistent change in temperature or humidity across these experiments (not shown), suggesting the lack of large-scale control applies to E3SM as well. Future work is needed to better understand the rain rate amount responses seen here, especially over land, and to identify the important scales (both spatial and temporal) as well as the role of convective organization."

[Figure]

Figure R1: (top row) temperature and (bottom row) specific humidity differences (control minus experiment) over (left) tropical ocean and (right) tropical land.

More discussion on such points would provide a lot of valuable information for researchers to decide which cloud decorrelation method to use in future studies.

Could the authors briefly comment on why the manuscript focuses on summer and winter averages, and not annual averages?

We found that there was a lot of cancellation in the temperature signal for the Clouds-off experiment between summer and winter, owing to the seasonality of the SWCRE versus LWCRE at mid- to high-latitudes. We therefore kept that split season structure throughout the manuscript to make sure we didn't miss other signals potentially masked by seasonal cancellation. We have clarified this point with the added line, "Because of the seasonally dependent role of clouds on the surface, we focus on the seasonal means instead of the annual mean. This focus on the seasonal means reduces the risk of missing important seasonal signals that cancel out in the annual mean."

Specific comments:

Line 190 – 198, Page 9-10:

While I believe that the details of the SST prescription do not substantially change the surface temperature response, I am not sure that the plot really shows this. Qualitatively, the pattern is indeed similar. But without repeating the same experiment with E3SM, it would be hard to make a good statement. My suggestion is to either add such an E3SM experiment or to remove Figure 4 from the manuscript.

While we haven't run out additional 36-year experiments to follow the same transient SST patterns used by the other models, we still want to keep Figure 4 as we believe it is valuable to establish that the prescription of SSTs is not important for deriving the same qualitative surface LWCRE warming response. The original text has been revised to make this point clearer and now reads, "The qualitative response of the surface LWCRE on the surface warming is consistent across all models (Figure 4) regardless of how the SSTs are prescribed. The large warming at high latitudes is not unique to E3SM."

Page 10:

How does surface in "surface locking" compare to ocean-covered areas? In one case, the surface fluxes are prescribed, while in the other case, sea surface temperatures are prescribed. Why did you decide to prescribe the surface energy budget and not directly temperature?

Both the surface fluxes and SSTs are prescribed in the Surface-locking experiment over oceans. We opted to prescribe the surface fluxes over land instead of prescribing the surface temperature for two reasons. First, we wanted to mirror what was done in Lau et al. (2019). Second, prescribing the surface fluxes allows us to control both heat and moisture fluxes. We have added the line, "By prescribing the surface fluxes, instead of, for example, prescribing the land surface temperature, we can control both the heat and moisture fluxes into the atmosphere."

Are substantial surface temperature anomalies a result of your relaxation of surface fluxes, that was in my understanding used to avoid numerical instability?

The surface temperature differences from the control are like those of the clouds-off ATM experiment because those are the flag settings used. Ideally, the surface-locking would further mute these differences even when using the clouds-off ATM flags, but that does not seem to be the case and we speculate it is the nudging that keeps the anomalies from decreasing in magnitude. Related to a question below about using surface locking with the clouds-off flag settings, we do find a reduction in the surface temperature differences.

[Figure]

Figure R2: (top row) JJAS and (bottom row) DJFM surface temperature differences (control minus experiment) for the (b + g) clouds-off, (c + h) clouds-off plus surface-locking, (d + i) clouds-off ATM, and (e + j) clouds-off ATM plus surface-locking experiments. Note that clouds-off ATM plus surface-locking is the same as the "surface-locking" experiment presented in the manuscript.

We have added the line, "Pairing the surface-locking with the Clouds-off flag settings (TTTT) reduces the surface temperature differences between control and experiment to magnitudes like those of the clouds-off ATM experiment while preserving the same qualitative warming and cooling pattern (not shown). Again, this result shows there can be value to surface locking, but its technical challenge makes it less attractive of an option compared to Clouds-off ATM."

Also, how is the surface locking method of Lau et al. 2019 that is used in this work different from the prescribed land surface temperature method by Ackerley and Dommenget, 2016 (10.5194/gmd-9-2077-2016)?

Our method provides the surface fluxes, while theirs prescribes surface temperature. The same way they output the surface temperature values from a control simulation with interactive land, we take our surface fluxes from a control. We also average the values to each time and day across all years, i.e., all years of Jan 1 at 00:00 are averaged together in both Ackerley and Dommenget (2016) and our study. We have added the lines, "Our method is also similar to that of Ackerley and Dommenget (2016) except for two key differences. First, as noted already, we prescribe surface fluxes instead of land surface temperatures. Second, as we will describe below, the fluxes are composited to smooth out the data while retaining the diurnal and seasonal cycles."

Would "surface temperature locking" be a more appropriate method to use when clouds are set to be invisible for radiation?

See answer above.

Page 12:

Figure 5 makes a good argument for the use of Clouds-off ATM instead of Clouds-off LW, although is referenced only in 1 paragraph in the text. It may be interesting to add Surface-locking results in it. Are the anomalies further improved in Surface-locking? If not, that would give another reason for arguing that the surface-locking method is not worth the effort.

The Surface-locking experiment produces similar differences from the control for both surface snowfall and snow depth over land as the Clouds-off ATM experiment. We have added in the Surface-locking experiment as a fourth column in Figure 5. Since the point of this figure is to examine the land response to the surface LWCRE, we have also masked out the ocean regions in the snowfall fluxes since the differences in snowfall are larger over ocean than land in the Clouds-off ATM and Surface-locking experiments and might be distracting to readers. We have added the following lines into the final paragraph of section 3.1.4 about the surface locking, "Similar results are found when examining the snowfall and snow water equivalent over land (Figure 5). The change in both snowfall and SWE between control and experiment agree for the clouds-off ATM and surface-locking experiments. The similarity of the results suggests little benefit to be gained for surface snow by prescribing the surface fluxes."

Page 14, Cloud-locking:

Could storing control simulation variables less frequently, e.g., every third radiation time step, along with some interpolation for model time steps between time steps with input data, be an alternative way to reduce the storage required by cloud locking?

It is not immediately clear that one could interpolate the cloud optical properties to a higher temporal frequency without unintended consequences. One could simply repeat the same clouds for intermediate radiative transfer calls, but this option has not been explored in E3SM. These impacts are interesting, but they would require future efforts to test practicality. We have added the line, "There may exist other strategies for lowering the storage overhead for cloud locking, such as reducing the frequency of storing the cloud optical properties. Middlemas et al. (2019) used 2-hourly cloud property inputs while keeping hourly radiative transfer calls and found that it was comparable to higher frequency cloud property inputs. Testing these strategies in E3SM is left to future efforts."

Page 17:

Figure 8 is only mentioned in one paragraph of the text. It could probably be referenced a few more times.

Figure 8 is also referenced in section 3.2.2 (the intro to the prescribed-RadHt experiment). We have also added in a reference to Figure 8 in section 4.1 as well as the conclusions section.

Page 16, line 356-360:

I guess this is done in the same way that SST is considered in AMIP simulations? (interpolation of monthly means to intermediate time steps). Mentioning this may help in understanding the implementation of Prescribed-RadHt and Prescribed-CRE.

Yes, the interpolation mirrors how SST is handled in AMIP simulations. We have added the parenthetical "(the same way the prescribed SSTs are handled)" to the end of the line, "The monthly climatological values are linearly interpolated to the current model time at each time step (the same way the prescribed SSTs are handled)." This was line 363-364 in the original manuscript.

Page 18, line 381:

Why is radiative heating prescribed only within the troposphere? This seems to make the implementation a bit more complex. Would the results be substantially different if monthly averaged heating were prescribed at all model levels?

We tested prescribing radiative heating at all model levels and found large temperature trends in the stratosphere. We found that these temperature trends led to a large jet shift in the Southern hemisphere (see below). Concerned how this might impact circulation metrics like the jet latitude, we opted to allow the stratosphere to use the online radiative heating rates and the temperature trends went away. We have added the above text to the manuscript as the lines, "We tested prescribing radiative heating at all model levels and found large temperature trends in the stratosphere. We found that these temperature trends led to a large jet shift in the Southern hemisphere (not shown). Concerned how this might impact circulation metrics for the jet, we opted to allow the stratosphere to use the online radiative heating rates and the stratospheric temperature trends went away."

[Figure]

Figure R3: Zonal mean zonal winds for the control (contours) and their differences (control minus experiment; shading). The lower right panel shows a test version of the prescribed-RadHt experiment that prescribed the radiative heating at all levels. The other panels show the covariance denial experiments used in the manuscript for reference.

Page 24, lines 499-511

More reference to Figures 13 and 14 may help get the message across. The BSISO explanation is very detailed and could be left for a follow-up publication.

Figures 13 and 14 refer to the complete radiation denial experiment types, while this paragraph discusses the covariance denial experiment types. We assume the reviewer meant more references to Figures 12 and 15, which we have added to the text. We have kept in the BSISO explanation since we believe it provides a useful example of future research directions based on these types of experiments.

Page 28, lines 584-594:

Are prescribed-RadHt and prescribed-CRE really good enough to study the role of CRE on circulation?

We agree this was a vague statement, as "good enough" has not been defined in this case. As pointed out by the other reviewer, internal variability may be confounding how we interpret the comparison of the three covariance denial experiments. Future work is needed to better understand these differences and we are planning to work on that. What we can say from this work is that the results cannot rule out these alternative methods as viable ones, making them a useful target for future research efforts. Given that more work is needed to understand the new alternatives, we have added in our recommendation for using the cloud-locking experiment type when resources are limited.

We have rewritten this paragraph to now read, "We have also demonstrated two CRE-circulation decorrelating experiment alternatives to the cloud-locking design: the prescribed-RadHt and prescribed-CRE. While not identical, all three do little to disrupt the mean circulation (Figure 8) and have minor influence on the surface temperatures (Figure 9). There is agreement in the precipitation response (Figure 15) and agreement in the sign of the change across these three experiment types for the responses in precipitation and the Hadley circulation within the Southern Hemisphere (Figure 16). There is disagreement in the sign of the change in the Northern Hemisphere and for the jet metrics, both of which may be influence by internal variability. Future work is needed to disentangle these differences between the three experiment types and determine where the differences are robust and what physical differences give rise to them. There is also substantial disagreement in the role of CRE-circulation covariations for the rain rate distributions shown in Figure 17, which requires future work to understand.

The prescribed-RadHt and prescribed-CRE experiments are of interest as alternatives to cloud locking because their input fields can more readily be taken from other models, reanalyses, or observations to be used to quantify the role of CRE biases on the circulation and water cycle. Future work is needed to test whether monthly data is the ideal frequency for the prescribed-RadHt and prescribed-CRE experiments or whether shorter timescale variability is needed. If monthly data can be shown to be sufficient, then the prescribed-RadHt and prescribed-CRE experiment types will continue to be sufficiently less data intensive than cloud-locking, making them an appealing alternative for high resolution modeling or for examining interannual or interdecadal variability associated with climate modes like El Nino Southern Oscillation or the Indian Ocean Dipole. At this time, however, we recommend the cloud-locking experiment when a covariance denial experiment type is needed since it is better understood and is already being used by the community in other models."

Could you, based on your simulations, make more detailed statements/suggestions/guidelines about which method is most appropriate for a particular scientific question?

In the conclusions, we recommend the clouds-off ATM experiments for understanding the role of mean ACRE on the circulation and climate (also addressed in the comment below). For the covariance denial experiment types, we recommend the cloud-locking owing to its widespread use by the community and since it is better understood at this time (see preceding comment as well).

Conclusions:

Your work points at Clouds-off ATM as the clear winner of the COOKIE-style experiments. If so, this should be stated even more clearly in the conclusions.

In the original manuscript we state, "When resources limit the number of experiments that can be performed, we recommend using the clouds-off ATM experiment for studies interested in better understanding the impact of ACRE on the present-day circulation."  To make this point even clearer, we add the following line immediately after that one, "Among the complete cloud radiation denial experiments (clouds-off, clouds-off LW, clouds-off ATM, and surface-locking) the clouds-off ATM is the ideal experiment type both in terms of its simplicity to use and its avoidance of potentially problematic surface temperature changes."

Data and code:

In which E3SM branch on github can the code be found?

In Appendix B, the branches are listed for the v1 and v2 codes as part of the step-by-step guide for checking out the right code.  While this was not exactly the issue raised by the reviewer, we have updated the instructions for getting the code from github to better match current best practices.  These changes can be found in Appendix B under "Step 1".  The lines for checking out the v1 repository now read:

```
git clone git@github.com:beharrop/E3SM.git

cd ./E3SM

git checkout beharrop/atm/cre_experiments

git submodule update –init
```

and the lines for checking out the v2 repository now read:

```
git clone git@github.com:beharrop/E3SM.git

cd ./E3SM

git checkout beharrop/atm/cre_experiments_v2

git submodule update --init
```

Could you add a branch-specific readme file with some information about the changes made (e.g. short summary, with a link to this manuscript), link to the manuscript(s) where the specific code was used?

Yes, this has been added to github (https://github.com/beharrop/E3SM/tree/beharrop/atm/cre_experiments).

Best regards,

Blaž Gasparini with comments from other members of the Climate Dynamics and Modeling team at the University of Vienna

**Reviewer #2**

This manuscript provides a detailed description of the experimental design of seven new cloud-radiation denial experiments using the E3SM model, which remove either the mean atmospheric cloud radiative effect (COOKIE/complete cloud-radiation denial experiments) or the covariance between the atmospheric cloud radiative effect and circulation (cloud locking/decorrelating experiments). These experiments are designed to explore the impact of the cloud radiative effect on various aspects of Earth's climate system. The manuscript details the technical descriptions of how to set up and run these experiments, as well as providing some sample results.

Overall, this manuscript provides a nice summary of these cloud-radiation denial methodologies in the E3SM model. A particular strength of the manuscript is a detailed methodological comparison of slightly different implementations of the COOKIE and cloud locking methodologies, which is something that has been absent from the literature and a very much needed contribution to better inform future studies in this area. I have a number of minor comments below, which are suggested to improve the clarity of the manuscript. In particular, I would encourage the authors to be clearer in describing variable names and namelist properties that are specific to E3SM, as this will broaden the reach of this study from E3SM users to other scientists who may wish to apply this methodology in other models.

We appreciate the reviewer's comments and suggestions for improvement. There are many great suggestions for improving the language and clarifying the descriptions, and we will work to address each of them.

Minor Comments:

Lines 44-45, 82-83, 316–323, and hereafter: I think better distinction needs to be made throughout this manuscript between cloud locking experiments for which the control simulation is a different climate (such as is often done in investigating the role of CRE in the climate change response; e.g., Ceppi and Hartmann 2016; Albern et al. 2019), versus cloud locking experiments for which the control simulation is the same climate (in this case, the CRE climatology remains identical, but the covariance between CRE and circulation is decoupled; e.g., Rädel et al. 2016; Grise et al. 2019). In the former (climate change) case, the mean CRE does not necessarily remain the same. For example, a control run (T0) could be run with clouds either prescribed to the present-day (C0) or warmer climates (C1). In this case, the mean CRE is dependent on the climate to which the clouds are locked (C0 or C1). In the latter case, the CRE is always locked to C0, but the individual years are scrambled so that the clouds do not co-vary with the circulation features.

The reviewer brings up a useful point. T0C0 refers to the baseline cloud-locking experiment that is compared to the free-running "control" simulation. This is probably most pertinent in the circulation metrics discussion since the "true" response is the difference between the control +4K and control

present-day, while the "total" cloud-locking response is the T1C1 – T0C0, or in other words, cloud-locking +4K SSTs and clouds minus cloud-locking present-day SSTs and clouds.  We have added the following lines to section 3.2.1 where we introduce the cloud-locking experiment, "Note that in the above, T0C0 is the same experiment as the "cloud-locking" experiment shown when comparing to the free running control.  Whenever we compare the "control" and "cloud-locking" runs in this work, both experiments use present-day SSTs and clouds, only the clouds are no longer correlated with the circulation in the "cloud-locking" run."

We have changed the second paragraph of section 4.2 in response to this and a comment raised by the reviewer (see response below).

Lines 72-73: It would be good to specify the length of all model runs used in this manuscript (i.e., how many years is each run in Table 1?).

We have added a column to Table 1 to denote which years of the 11 year simulations are used for analysis (they all use years 2-11).

Lines 207–214: More explanation of these results is needed.  Why is there an increase in snowfall for both runs, even though the changes in winter surface temperature in these regions are very different between the runs (Fig. 2)?

In the regions of increased snowfall total precipitation increases as well, which is hard to see in Figure 12 owing to the colorbar so we have attached a modified version of Figure 5 here (total precipitation is the third row).  Despite the large temperature responses, the surface temperature in the Arctic still gets cold enough in all of these runs to allow for snow, so the increased precipitation drives an increase in snowfall.  We have modified the line, "For example, Figure 5 (top row) shows that ACRE increases the amount of snowfall in the Arctic (total precipitation also increases; not shown)," to now read, "For example, Figure 5 (top row) shows that ACRE increases the amount of snowfall in the Arctic.  A similar increase in total precipitation occurs over the same Arctic area (not shown).  Despite the large temperature changes shown in Figure 2, the temperatures still get sufficiently cold to allow for snow, such that the increased precipitation from ACRE leads to an increase in snowfall for all experiments," to add clarity.

[Figure]

Figure R4: (top row) snowfall, (middle row) snow water equivalent on land, and (bottom row) total precipitation.

Lines 251-255: In this paragraph, you also need to discuss what the presc_sfc_flux_cycle_yr and pertlim namelist settings mean.

We thank the reviewer for pointing this out. The workflow figure for the surface-locking was an old version of that figure. The `presc_sfc_flux_cycle_yr' argument was meant to be an optional argument, but was later removed, so it wasn't included in the text. We have updated the manuscript with the correct version of this figure. We have included the description of `pertlim' into the text as, "the bounds of the noise to add to the temperature initial conditions to force the weather to diverge from the control experiment (pertlim)."

Line 347 (Equation 2): It's more conventional to calculate the mass streamfunction integrating downward from the top of the atmosphere, rather than upward from the surface. See Chapter 6 of Hartmann's Global Physical Climatology textbook. This prevents issues with model representation of surface pressure from impacting the streamfunction calculation over the depth of the troposphere.

This was a typo in the manuscript. The code used to do this calculation integrates from 0 to $p$ as expected. We have corrected equation 2.

Line 385: It's good to note which vertical level indices correspond to 25-80 hPa, as level indices are listed in the p_radht_coefs variable given on lines 392-398.

We added in that this transition layer corresponds to level indices 16 through 22. The line now reads, "The transition zone, where the radiative heating is a weighted combination of prescribed and online computed heating occurs roughly around pressure levels 25-80~ hPa (levels 16-22 in EAM)."

Lines 386-389: Why is cpair multiplied by qrs_input and qrl_input, but not by qrs and qrl? I see the note on Line 425, but you may need to make a similar note here.

We have clarified that cpair is already included in qrs and qrl at this point in the model with the line, "Note that the specific heat of dry air is already included in the qrs and qrl terms, but not in qrs_input or qrl_input."

Lines 421-425: Why do the fsnt, flnt, fsns, and flns cloud variables have dimensions of (i,1), but the clear-sky variables have dimensions of (i)? (Note: I now see why after reading Appendix A, but it may be good to refer readers here.)

We have added in the line, "Note that the data read in from file have a singleton level dimension owing to a bug in how the model reads in transient 2D data (more discussion in Appendix A)."

Lines 443-444: It would be good to define q here as well.

We have added, "and $q$ is the specific humidity" here.

Lines 482–484: See my earlier comment about the length of the runs. If the dynamical responses to ACRE are small relative to the magnitude of internal variability in these regions, you would need a relatively lengthy model run to be able to discern the signal from the noise. So, if the runs are not long enough, some of the discrepancy here could be internal variability, rather than being physically meaningful.

We have addressed this comment in our response to reviewer #1 above.

Lines 499-511, Section 4.3: See first comment above. Here, if I understand correctly, it appears the authors are locking the clouds to the clouds of the same climate, but with the clouds and circulation de-coupled (similar to Rädel et al. 2016). Again, better clarity is needed when describing what is done in these locking-type simulations, as the discussion in Section 3.2 is focused almost entirely on describing the cloud locking methodology for the climate change response (as shown in Fig. 16), and not the type of experiments shown in Figs. 15 and 17.

We have added in these details in our response to the previous comment on this topic.

Line 511: See also Benedict et al. (2020), who examined the role of cloud-circulation interactions on modes of tropical intraseasonal variability.

Thank you for the suggestion. We have added in the line to section 4.1, "Benedict et al. (2020) found decorrelating CRE and circulation led to a weakening of the MJO amplitude (though it increased the strength of Kelvin waves)."

Section 5.2: The text needs to be clearer that this entire section is discussing the circulation response to warming.

We have changed the opening line to read, "Next, as noted in section 3.2.1, we use the same decomposition as Voigt and Albern (2019, their equation 5) to examine the cloud circulation feedbacks to the general circulation changes under a +4K warming scenario."

Lines 528-530: I find it difficult to follow what the authors are doing here, even after referring back to the Voigt and Albern (2019) paper. I would suggest showing the equation for the SST response in the manuscript.

We have opted to move some of the discussion of the factorial experiments and equation 1 from section 3.2.1 to section 4.2 where it is used (lines 324-329 of original manuscript). We have left in section 3.2.1 a simplified discussion of the factorial experiments to motivate our decision for free running water vapor in the current experiments. The new text in section 3.2.1 now reads, "In order to compute the role of clouds on differences resulting from climate changes, cloud-locking experiments typically include running factorial experiments where SSTs and prescribed cloud properties are toggled for multiple climate states. For example, if `0' denotes the present-day climate, `1' denotes the +4K warming climate, `T' denotes the SST choice, and `C' denotes the cloud property choice, then the four experiments would be T0C0, T0C1, T1C0, and T1C1. These experiments can then be mixed-and-matched to extract the cloud response, SST response, and residual (more discussion on this in section 4.2). As noted by Voigt and Albern (2019), this method can also include locking water vapor (see Voigt and Shaw, 2015). If water vapor is added to the factorial experiment design, computing all of the terms requires combining eight unique simulations (see equation 1 of Voigt and Albern, 2019). If it is assumed that water vapor must be consistent with SSTs to give credible simulations (either by locking water vapor to corresponding SSTs or by allowing for free-running water vapor), then the number of simulations required to compute the cloud response can be reduced to four. Voigt and Albern (2019) show (their Figure 1) that most features of the climate system are reliably reproduced regardless of the choices surrounding water vapor. As a result, we opt to allow for free-running water vapor in the cloud-locking simulations performed for this study, and make use of only four experiments to determine cloud responses using cloud-locking."

In section 4.2, we have rewritten the paragraph (original lines 524-530) to read, "Figure 16 shows each of these metrics for the annual mean of each hemisphere. Voigt and Albern (2019) note that assessing the cloud impact is only relevant when the residual term is less than one third of the true change (as measured using the control +4K and present-day climatology runs).

   True response = (Control +4K - Control)

The true value is marked as a solid black horizontal line, and one third of its value is provided as a gray, dashed line. Note that the ``true'' response is distinct from the ``total'' response measured as T1C1 – T0C0. Here, T1C1 refers to the cloud-locking experiment where the +4K SSTs and clouds are used, and T0C0 refers to the cloud-locking experiment where the present-day SSTs and clouds are used. In both T1C1 and T0C0, the clouds are decorrelated from the circulation. In the true response, using the control with present-day and +4K SSTs, the clouds are still correlated with the circulation. The residual terms for each of the cloud-locking, prescribed-RadHt, and prescribed-CRE experiments are given in the non-shaded bars, and their cloud contribution terms are provided in the shaded bars. The residual is computed as in Voigt and Albern (2019, their equation 3), which is

   Residual response = (True response – Total response)

= (Control +4K - Control) - (T1C1 - T0C0)

Following the framework of Voigt and Albern (2019), the cloud and SST responses are as follows.

Cloud response = 1/2 ((T0C1 - T0C0) + (T1C1 - T1C0))

SST response = 1/2 ((T1C1 - T0C1) + (T1C0 - T0C0))

Again, Figure 16 shows the residual (Equation 10) and cloud (Equation 11) response terms. Where these terms exceed one third of the true response (Equation 8) is where we expect the term to be a meaningful contribution, based on Voigt and Albern (2019).

Lines 545-548: I don't understand what the authors are arguing in this sentence. The radiative heating and CRE locking runs also show a poleward shift in the SH Hadley cell edge, consistent with the poleward shift in P-E = 0.

We have revised this line for clarity to read, "The expansion of the SH subtropical dry zone edge, Hadley cell edge, and jet latitude are all consistent and robust for only the cloud-locking experiment, suggesting the cloud-locking experiment type may have an advantage over the prescribed-RadHt and prescribed-CRE experiments, but more work is needed to understand where and how these experiments differ for these metrics."

Lines 585-586: I would disagree with this assessment. For example, there are fairly large differences in the precipitation distributions in Fig. 17 and in the circulation responses to climate change shown in Fig. 16. While these new methodologies are intriguing, I think the verdict is still out as to whether they can be used to replace the cloud locking methodology.

We have clarified this point in our response to reviewer #1 above.

Figure 5: Statistical significance should be noted on this figure. It seems important to know whether the differences between clouds-off-LW and clouds-off-ATM are robust, or relatively small compared to internal variability.

We have added stippling to Figure 5 to demonstrate the changes between the control and experiments are significant at the 95% level.

Figure 8: Again, it's important to know where the responses from the control are statistically significant.

We have also added in stippling to Figure 8.

Figures 13-14: Are these calculations for annually averaged P-E or summertime P-E? Also, statistical significance should be noted in Fig. 13, as in earlier figures.

They are for local summer (JJAS in N. Hemisphere, DJFM for S. Hemisphere), and this clarification has been made in the captions for Figure 13 and 14. We have added stippling to Figure 13 as well. The added line for the captions reads, "All values are for local summer (JJAS in Northern Hemisphere; DJFM in Southern Hemisphere)."

Typos:

Lines 7-8: necessary to implement Corrected

Line 113: Missing ) Corrected

Line 156: Do you mean atmospheric heating rates or atmospheric layer heating rates? Corrected to atmospheric layer heating rates

Line 228: I think lwup should be FLUS to be consistent with Line 241. In the E3SM code, both "lwup" and "FLUS" are used to denote upwelling surface longwave radiation. FLUS is the variable name used for the model history field to be consistent with the naming conventions of the other radiative fluxes and lwup is the variable name used by the coupler (upwelling longwave fluxes are computed in the surface models and passed as input to EAM for use in the radiative transfer code). While these values are the same, since the surface flux overwriting process changes the coupler variables used as input to EAM, we use lwup in this instance. We have added the clarifying line, "Note that while "FLUS" is the model history variable name, "lwup" is the variable name used by the coupler that we overwrite and is used in the pseudocode above."

Lines 382: based --- this seems like an incomplete sentence, based on what? It is a typo. "Based" should not be there at all. We have removed it.

References:

Albern, N., Voigt, A., & Pinto, J. G. (2019). Cloud-radiative impact on the regional responses of the midlatitude jet streams and storm tracks to global warming. Journal of Advances in Modeling Earth Systems, 11, 1940–1958. https://doi.org/10.1029/2018MS001592

Benedict, J. J., Medeiros, B., Clement, A. C., & Olson, J. (2020). Investigating the role of cloud-radiation interactions in subseasonal tropical disturbances. Geophysical Research Letters, 47, e2019GL086817. https://doi.org/10.1029/2019GL086817

Ceppi, P., and D. L. Hartmann, 2016: Clouds and the Atmospheric Circulation Response to Warming. J. Climate, 29, 783–799, https://doi.org/10.1175/JCLI-D-15-0394.1.

Grise, K. M., Medeiros, B., Benedict, J. J., & Olson, J. G. (2019). Investigating the influence of cloud radiative effects on the extratropical storm tracks. Geophysical Research Letters, 46, 7700–7707. https://doi.org/10.1029/2019GL083542

Rädel, G., Mauritsen, T., Stevens, B., Dommengat, D., Matei, D., Bellomo, K., & Clement, A. (2016). Amplification of El Niño by cloud longwave coupling to atmospheric circulation. Nature Geoscience, 9(2), 106–110. https://doi.org/10.1038/ngeo2630